# Generalization of GANs and overparameterized models under Lipschitz continuity

## Abstract

Generative adversarial networks (GANs) are so complex that the existing learning theories do not provide a satisfactory explanation for why GANs have great success in practice. The same situation also remains largely open for deep neural networks. To fill this gap, we introduce a Lipschitz theory to analyze generalization. We demonstrate its simplicity by showing generalization and consistency of overparameterized neural networks. We then use this theory to derive Lipschitz-based generalization bounds for GANs. Our bounds show that penalizing the Lipschitz constant of the GAN loss can improve generalization. This result answers the long mystery of why the popular use of Lipschitz constraint for GANs often leads to great success, empirically without a solid theory. Finally but surprisingly, we show that, when using Dropout or spectral normalization, both *truly deep* neural networks and GANs can generalize well without the curse of dimensionality.

## 1 Introduction

In *Generative Adversarial Networks* (GANs) (Goodfellow et al., 2014), we want to train a discriminator $D$ and a generator $G$ by solving the following problem:

$$\min_{G} \max_{D} \mathbb{E}_{x \sim P_d} \log(D(x)) + \mathbb{E}_{z \sim P_z} \log(1 - D(G(z))) \tag{1}$$

where $P_d$ is a data distribution that generates real data, and $P_z$ is some noise distribution. $G$ is a mapping that maps a noise $z$ to a point in the data space. After training, $G$ can be used to generate novel but realistic data.

Since its introduction (Goodfellow et al., 2014), a significant progress has been made for developing GANs and for interesting applications (Hong et al., 2019). Some recent works (Brock et al., 2019; Zhang et al., 2019; Karras et al., 2020b) can train a generator that produces synthetic images of extremely high quality. To explain those success, one popular way is to analyze generalization of the trained players. There are many existing theories (Mohri et al., 2018) for analyzing generalization. However, they suffer from various difficulties since the training problem of GANs is unsupervised in nature and contains two players competing against each other. Such a nature is entirely different from traditional learning problems. Neural distance (Arora et al., 2017) was introduced for analyzing generalization of GANs. One major limitation of existing distance-based bounds (Arora et al., 2017; Zhang et al., 2018; Jiang et al., 2019; Husain et al., 2019) is the strong dependence on the capacity of the family, which defines the distance between two distributions, sometimes leading to trivial bounds. This limitation prevents us from fullly understanding and identifying the key factors that contribute to the generalization of GANs. For example, it has long been theoretically unclear *why Lipschitz constraint empirically can lead to great success in GANs?*

The standard learning theories suffer from various difficulties when analyzing overparameterized neural networks (NNs). For example, Radermacher-based bounds (Bartlett & Mendelson, 2002; Golowich et al., 2020) can be trivial (Zhang et al., 2021); algorithmic stability (Shalev-Shwartz et al., 2010) and robustness (Xu & Mannor, 2012) may not be directly used since instability is the well-known issue when training GANs (Salimans et al., 2016; Arjovsky & Bottou, 2017; Xu et al., 2020). Those examples are among the reasons for why the theoretical study of modern deep learning is still in its infancy (Fang et al., 2021). Although some studies show generalization for shallow networks with at most one hidden layer (Arora et al., 2021; Mianjy & Arora, 2020; Mou et al., 2018; Kuzborskij & Szepesvári, 2021; Ji et al., 2021; Hu et al., 2021; Jacot et al., 2018), it has

long been theoretically unknown *why deeper NNs can generalize better?* Many great successes of deep learning often need huge datasets, but it has long been theoretically unknown *whether or not the generalization of deep NNs suffers from the curse of dimensionality?*

This work has the following contributions:

▷ We introduce a *Lipschitz theory* to analyze generalization of a learned function. This theory is surprizingly simple to analyze various complex models in general settings (including supervised, unsupervised, and adversarial settings).

▷ We show that Dropout or spectrally-normalized neural networks avoid the curse of dimensionality. The number of layers required to ensure good generalization is *logarithmic* in sample size. It also suggests that deeper NNs can generalize better. We further show consistency and indentify a *sufficient condition* to guarantee high performance of DNNs. These results provide a significant step toward answering the open theoretical issues of deep learning (Zhang et al., 2021; Fang et al., 2021).

▷ Using Lipschitz theory, we provide a comprehensive analysis on generalization of GANs which resolves the two open challenges in the GAN community: **(i)** Our bounds apply to any particular $D$ or $G$, and hence overcome the major limitation of existing works; In particular, for the first time in the literature, we show that Dropout and spectral normalization can help GANs to avoid the curse of dimensionality. **(ii)** Lipschitz constraint is used popularly through various ways including gradient penalty (Gulrajani et al., 2017), spectral normalization (Miyato et al., 2018), dropout, and data augmentation. Our analysis provides an *unified explanation* for why imposing a Lipschitz constraint can help GANs to generalize well in practice.

*Organization:* We will review related work in the next section. Section 3 presents the theory connecting Lipschitz continuity with generalization, and some analyses about deep neural networks. In Section 4, we analyze the generalization of GANs in various aspects. Section 5 concludes the paper.

## 2 RELATED WORK

*Generalization in GANs:* There are few efforts to analyze the generalization for GANs using the notion of *neural distance*, $d_{\mathcal{D}}(P_d, P_g)$, which is the distance between two distributions $(P_d, P_g)$, where $\mathcal{D}$ is the discriminator family.[1] Arora et al. (2017) analyze generalization by bounding the quantity $|d_{\mathcal{D}}(P_d, P_g) - d_{\mathcal{D}}(\widehat{P}_d, \widehat{P}_g)|$, where $(\widehat{P}_d, \widehat{P}_g)$ are empirical versions of $(P_d, P_g)$. For a suitable choice of the loss $V(P_d, P_z, D, G)$ in GANs, we can write $|d_{\mathcal{D}}(P_d, P_g) - d_{\mathcal{D}}(\widehat{P}_d, \widehat{P}_g)| = |\max_{D \in \mathcal{D}} V(P_d, P_z, D, G) - \max_{D \in \mathcal{D}} V(\widehat{P}_d, \widehat{P}_z, D, G)|$, where $P_g$ is the induced distribution by putting samples from $P_z$ through generator $G$. Both Arora et al. (2017) and Husain et al. (2019) analyze $|\max_{D \in \mathcal{D}} V(P_d, P_z, D, G_o) - \max_{D \in \mathcal{D}} V(\widehat{P}_d, \widehat{P}_z, D, G_o)|$ to see generalization of a trained $G_o$, while (Zhang et al., 2018; Jiang et al., 2019) provide upper bounds for $|\max_{D \in \mathcal{D}} V(P_d, \widehat{P}_z, D, G) - \min_{G \in \mathcal{G}} \max_{D \in \mathcal{D}} V(P_d, P_z, D, G)|$. Note that those quantities of interest are non-standard in terms of learning theory.

A major limitation of those distance-based bounds (Arora et al., 2017; Zhang et al., 2018; Jiang et al., 2019; Husain et al., 2019) is the dependence on the notion of distance $d_{\mathcal{D}}(\cdot, \cdot)$ which relies on the best $D \in \mathcal{D}$ for measuring proximity between two distributions. The distance between two given distributions $(\mu, \nu)$ may be small even when the two are far away (Arora et al., 2017). This is because there exists a perfect discriminator $D$, whenever $\mu$ and $\nu$ do not have overlapping supports (Arjovsky & Bottou, 2017). In those cases, a distance-based bound may be trivial. As a result, existing distance-based bounds are insufficient to understand generalization of GANs.

Qi (2020) shows a generalization bound for their proposed Loss-Sensitive GAN. Nonetheless, it is nontrivial to make their bound to work with other GAN losses. Wu et al. (2019) show that the discriminator will generalize if the learning algorithm is differentially private. Their concept of *differential privacy* basically requires that the learned function will change negligibly if the training set slightly changes. Such a requirement is known as *algorithmic stability* (Xu et al., 2010) and is nontrivial to assure in practice. Note that this assumption cannot be satisfied for GANs since their training is well-known to be unstable in practice.

---

[1]In general, $\mathcal{D}$ can be replaced by another family of functions to define the neural distance. However, for the ease of comparison with our work, the discriminator family is used.

*Lipschitz continuity, stability, and generalization:* Lipschitz continuity naturally appears in the formulation of Wasserstein GAN (WGAN) (Arjovsky et al., 2017). It was then quickly recognized as a key to improve various GANs (Fedus et al., 2018; Lucic et al., 2018; Mescheder et al., 2018; Kurach et al., 2019; Jenni & Favaro, 2019; Wu et al., 2019; Zhou et al., 2019; Qi, 2020; Chu et al., 2020). Gradient penalty (Gulrajani et al., 2017) and spectral normalization (Miyato et al., 2018) are two popular techniques to constraint the Lipschitz continuity of $D$ or $G$ w.r.t their *inputs*. Some other works (Mescheder et al., 2017; Nagarajan & Kolter, 2017; Sanjabi et al., 2018; Nie & Patel, 2019) suggest to control the Lipschitz continuity of $D$ or $G$ w.r.t their *parameters*. Data augmentation is another way to control the Lipschitz constant of the loss, and is really beneficial for training GANs (Zhao et al., 2020a;b; Zhang et al., 2020; Tran et al., 2021). Those works empirically found that Lipschitz continuity can help improving stability and generalization of GANs. However, it has long been a mystery of why imposing a Lipschitz constraint can help GANs to generalize well. This work provides an unified explanation.

## 3 LIPSCHITZ CONTINUITY AND GENERALIZATION

In this section, we will present the main theory that connects Lipschitz continuity and generalization. We then discuss why deep neural networks can avoid the curse of dimensionality, and why deeper networks may generalize better.

*Notations:* Consider a *learning problem* specified by a function/hypothesis class $\mathcal{H}$, an instance set $\mathcal{X}$ with diameter at most $B$, and a loss function $f : \mathcal{H} \times \mathcal{X} \to \mathbb{R}$ which is bounded by a constant $C$. Given a distribution $P_x$ defined on $\mathcal{X}$, the quality of a function $h(x)$ is measured by its *expected loss* $F(P_x, h) = \mathbb{E}_{x \sim P_x}[f(h, x)]$. Since $P_x$ is unknown, we need to rely on a finite training sample $\boldsymbol{S} = \{x_1, ..., x_m\} \subset \mathcal{X}$ and often work with the empirical loss $F(\widehat{P}_x, h) = \mathbb{E}_{x \sim \widehat{P}_x}[f(h, x)] = \frac{1}{m} \sum_{x \in \boldsymbol{S}} f(h, x)$, where $\widehat{P}_x$ is the empirical distribution defined on $\boldsymbol{S}$. A *learning algorithm* $\mathcal{A}$ will pick a function $h_m \in \mathcal{H}$ based on input $\boldsymbol{S}$, i.e., $h_m = \mathcal{A}(\mathcal{H}, \boldsymbol{S})$.

We first establish the following result whose proof appears in Appendix A.

**Theorem 1** (Lipschitz continuity $\Rightarrow$ Generalization). *If a loss $f(h, x)$ is $L$-Lipschitz continuous w.r.t input $x$ in a compact set $\mathcal{X} \subset \mathbb{R}^{n_x}$, for any $h \in \mathcal{H}$, and $\widehat{P}_x$ is the empirical distribution defined from $m$ i.i.d. samples from distribution $P_x$, then $\sup_{h \in \mathcal{H}} |F(P_x, h) - F(\widehat{P}_x, h)|$ is upper-bounded by*

*1. $L\lambda + C\sqrt{(\lceil B^{n_x} \lambda^{-n_x} \rceil \log 4 - 2 \log \delta)/m}$ with probability at least $1 - \delta$, for any constants $\delta \in (0, 1]$ and $\lambda \in (0, B]$.*
*2. $(LB + 2C)m^{-\alpha/n_x}$ with probability at least $1 - 2\exp(-0.5m^\alpha)$, for any $\alpha \le n_x/(2 + n_x)$.*

The assumption about Lipschitzness is natural. When learning a bounded function (e.g. a classifier), the assumption will satisfy if choosing a suitable loss (e.g. square loss, hinge loss, ramp loss, logistic loss, tangent loss, pinball loss) and $\mathcal{H}$ which has Lipschitz members with bounded ouputs. Cross-entropy loss can satisfy if we require every $h \in \mathcal{H}$ to have outputs belonging to a closed interval in (0,1) or use label smoothing. Some recent works (Miyato et al., 2018; Gouk et al., 2021; Pauli et al., 2021) propose to put a penalty on the Lipschitz constant of $h$ only. However, leaving open the Lipschitzness of $f$ w.r.t $h$ may not ensure a small Lipschitz constant of the loss.

This theorem tells that Lipschitz continuity is the key to ensure a function to generalize. Its generalization bounds can be better as the Lipschitz constant of the loss decreases. Note that there is a tradeoff between the Lipschitz constant and the expected loss $F(P_x, h)$ of the learnt function. A smaller $L$ means that both $f$ and $h$ are getting simpler and flatter, and hence may increase $F(P_x, h)$. In contrast, a decrease of $F(P_x, h)$ may require $h$ to be more complex and hence may increase $L$.

Theorem 1 presents generalization bounds, in a general setting, which suffer from the curse of dimensionality. This limitation is common for any other approaches without further assumptions (Bach, 2017). For some special classes, we can overcome this limitation as discussed next.

### 3.1 DEEP NEURAL NETWORKS THAT AVOID THE CURSE OF DIMENSIONALITY

We consider the two families of neural networks: one with bounded spectral norms for the weight matrices, and the other with Dropout. The following theorem whose proof appears in Appendix A.1 provides sharp bounds for the Lipschitz constant.

**Theorem 2.** *Let fixed activation functions $(\sigma_1, \ldots, \sigma_K)$, where $\sigma_i$ is $\rho_i$-Lipschitz continuous. Let $h_{\mathcal{W}}(x) := \sigma_K(W_K \sigma_{K-1}(W_{K-1} \ldots \sigma_1(W_1 x) \ldots))$ be the neural network associated with weight matrices $(W_1, \ldots, W_K)$, and $L_h$ be the Lipschitz constant of $h$. Let the bounds $(s_1, \ldots, s_K)$ and $(b_1, \ldots, b_K)$ be given.*

***Spectrally-normalized networks (SN-DNN):*** *Let $\mathcal{H}_{sn} = \{h_{\mathcal{W}} : \mathcal{W} = (W_1, \ldots, W_K), \|W_i\|_\sigma \leq s_i\}$, where $\|\cdot\|_\sigma$ is the spectral norm. Then $\forall h \in \mathcal{H}_{sn}$, $L_h \leq \prod_{k=1}^{K} \rho_k s_k$.*

***Dropout DNN:*** *Let $\mathcal{H}_{dr} = \{h_{\mathcal{W},q} : h_{\mathcal{W},q} = DrT(h_{\mathcal{W}}, q), \|W_i\|_F \leq b_i\}$, where $DrT$ is the usual dropout training (Srivastava et al., 2014) with drop rate $q$ for network $h_{\mathcal{W}}$, and $\|\cdot\|_F$ is the Frobenius norm. Then $\forall h \in \mathcal{H}_{dr}$, $L_h \leq q^K \prod_{k=1}^{K} \rho_k b_k$.*

Most popular activation functions (e.g., ReLU, Leaky ReLU, Tanh, SmoothReLU, Sigmoid, and Softmax) have small Lipschitz constants ($\rho_k \leq 1$). This theorem suggests that the Lipschitz constant can be exponentially small as a neural network is deep (large $K$) and uses Dropout at each layer, since $q < 1$ is a popular choice in practice. On the other hand, the Lipschitz constant will be small if we control the spectral norms of weight matrices, e.g. by using spectral normalization. The Lipschitz constant will be exponentially smaller as the neural network is deeper and the spectral norm at each layer is smaller than 1. This case often happens as observed by Miyato et al. (2018).

The generalization of SN-DNNs and Dropout DNNs can be seen by combining Theorems 1 and 2. One can observe that, for the same norm bound on weight matrices, a network with smaller Lipschitz constant can provide a better bound. An interesting implication from Theorem 2 is that deeper networks (larger $K$) will have smaller Lipschitz constants and hence lead to better generalization bounds. This answers the second question of Section 1.

A trivial combination of Theorems 1 and 2 will result in a bound of $O(m^{-1/n_x})$ which suffers from the curse of dimensionality. The following theorem shows stronger results in Appendix A.1.

**Theorem 3** (Generalization of DNNs). *Given the assumptions in Theorems 1 and 2, let $L_f$ be the Lipschitz constant of the loss $f(h, x)$ w.r.t $h$, and $\delta \in (0, 1]$ be any given constant.*

*1. **SN-DNNs:** assume that there exist $p \in (0, 1)$ and constant $C_{sn}$ such that $C_{sn} p^K \geq \prod_{k=1}^{K} \rho_k s_k$. If the number of layers $K \geq -\frac{1}{2} \log_p m$, then the following holds with probability at least $1 - \delta$:*

$$\sup_{h \in \mathcal{H}_{sn}} |F(P_x, h) - F(\widehat{P}_x, h)| \leq \left(C_{sn} L_f B + C\sqrt{\log 4 - 2\log \delta}\right) m^{-0.5}$$

*2. **Dropout DNNs:** For $\mathcal{H}_{dr}$ with drop rate $q \in (0, 1)$, let $C_{dr} = \prod_{k=1}^{K} \rho_k b_k$. If the number of layers $K \geq -\frac{1}{2} \log_q m$, then the following holds with probability at least $1 - \delta$:*

$$\sup_{h \in \mathcal{H}_{dr}} |F(P_x, h) - F(\widehat{P}_x, h)| \leq \left(C_{dr} L_f B + C\sqrt{\log 4 - 2\log \delta}\right) m^{-0.5}$$

The assumption of $K \geq -\frac{1}{2} \log_q m$ is naturally met in practice. For example, when training from 1.2M images, constant $q = 0.5$ requires $K \geq 10$, and $q = 0.1$ requires $K \geq 3$. Note that Alexnet (Krizhevsky et al., 2012) has 8 layers and the generator of StyleGAN (Karras et al., 2021) has 18 layers. The assumption about SN-DNNs can be satisfied when choosing activations with Lipschitz constant $\rho_k < 1$ or ensuring the spectral bound $s_k < 1$ at any layer $k$. As mentioned before, such conditions are often satisfied in practice (Miyato et al., 2018) when using spectral normalization.

*Comparison with state-of-the-art:* Some recent studies (Bartlett et al., 2017; Neyshabur et al., 2018) provide generalization bounds for SN-DNNs for classification problems, using Radermacher complexity or PAC-Bayes. One major limitation of their works is that the sample size depends polynomially/exponentially on depth $K$. For SN-DNNs using ReLU, Golowich et al. (2020) improved the dependence to be linear in $K$ if provided assumptions comparable with ours. In another view, when fixing $m$, their results require $K = O(m)$ to get a meaningful generalization bound. This is impractical. In contrast, our result shows that it is sufficient to choose $K$ which is logarithmic in $m$. Another limitation of the bounds in (Bartlett et al., 2017; Neyshabur et al., 2018; Golowich et al., 2020) is the dependence on $1/\gamma$, where $\gamma$ is the margin of the classification problem. Note that practical data may have a very small margin or may be inseparable. Hence those bounds are really limited and inapplicable to inseparable cases. On the contrary, Theorem 3 holds in general settings, including inseparable classification and unsupervised problems.

Our result for Dropout DNNs holds in general settings including unsupervised learning. This is significant since state-of-the-art studies about Dropout (Arora et al., 2021; Mianjy & Arora, 2020;

Mou et al., 2018) obtain efficient bounds only for networks with no more than 3 layers and for supervised learning. To the best of our knowledge, this work is the first showing that Dropout can help DNNs avoid the curse of dimensionality in general settings.

## 3.2 CONSISTENCY OF OVERPARAMETERIZED MODELS

We have discussed generalization of a function by bounding the difference between the empirical and expected losses. In some situations, those bounds may not be enough to explain a high performance, since both losses may be large despite their small difference. Next we consider consistency (Shalev-Shwartz et al., 2010) to see the goodness of a function compared with the best in its family.

**Definition 1.** *A learning algorithm $\mathcal{A}$ is said to be **Consistent** with rate $\epsilon_{cons}(m)$ under distribution $P_x$ if for all $m$, $\mathbb{E}_{\boldsymbol{S} \sim P_x^m} |F(P_x, \mathcal{A}(\mathcal{H}, \boldsymbol{S})) - F(P_x, h^*)| \leq \epsilon_{cons}(m)$, where $\epsilon_{cons}(m)$ must satisfy $\epsilon_{cons}(m) \to 0$ as $m \to \infty$, $h^* = \arg\min_{h \in \mathcal{H}} F(P_x, h)$.*

Consistency says that, for any (but fixed) $m$, the learned function $h_m = \mathcal{A}(\mathcal{H}, \boldsymbol{S})$ is required to be (in expectation) close to the optimal $h^*$. The closeness is measured by $|F(P_x, h_m) - F(P_x, h^*)|$. By considering this quantity, optimization error will naturally appear. We first show the following observation in Appendix A.2.

**Lemma 1.** *Denote $h^* = \arg\min_{h \in \mathcal{H}} F(P_x, h)$ and $\widehat{P}_x$ is the empirical distribution defined from a sample $\boldsymbol{S}$ of size $m$. For any $h_o \in \mathcal{H}$, letting $\epsilon_o = F(\widehat{P}_x, h_o) - \min_{h \in \mathcal{H}} F(\widehat{P}_x, h)$, we have:*
$$|F(P_x, h_o) - F(P_x, h^*)| \leq \epsilon_o + 2\sup_{h \in \mathcal{H}} |F(P_x, h) - F(\widehat{P}_x, h)|$$

This lemma shows why the optimization error $\epsilon_o$ and capacity of family $\mathcal{H}$ control the goodness of a function. Combining Theorem 1 with Lemma 1 will lead to the following.

**Theorem 4** (General family). *Given the assumptions in Theorem 1, consider any function $h_o \in \mathcal{H}$. Let $h^* = \arg\min_{h \in \mathcal{H}} F(P_x, h)$, and $\epsilon_o = F(\widehat{P}_x, h_o) - \min_{h \in \mathcal{H}} F(\widehat{P}_x, h)$ be the optimization error of $h_o$ on a sample $\boldsymbol{S}$ of size $m$. For any $\alpha \leq n_x/(2 + n_x)$, with probability at least $1 - 2\exp(-0.5m^\alpha)$: $|F(P_x, h_o) - F(P_x, h^*)| \leq \epsilon_o + 2(LB + 2C)m^{-\alpha/n_x}$.*

**Corollary 1.** *Given the assumptions in Theorem 4, consider a learning algorithm $\mathcal{A}$ and family $\mathcal{H}$. $\mathcal{A}$ is consistent if, for any given sample $\boldsymbol{S}$ of size $m$, the learned function $h_o = \mathcal{A}(\mathcal{H}, \boldsymbol{S})$ has optimization error at most $\epsilon_o(m)$ which is a decreasing function of $m$, i.e., $\epsilon_o(m) \to 0$ as $m \to \infty$.*

The assumption about optimization error $\epsilon_o(m)$ is naturally satisfied when the training problem is convex. Indeed, it is well-known (Allen-Zhu, 2017; Schmidt et al., 2017) that gradient descent (GD) with $T$ iterations can find a solution with error $O(T^{-1})$ whereas stochastic gradient descent (SGD) with $T$ iterations can find a solution with error $O(T^{-0.5})$. Therefore, GD and SGD with $T = O(m)$ iterations will satisfy this assumption. Note that convex training problems appear in many traditional models (Hastie et al., 2017), e.g., linear regression, support vector machines, kernel regression.

For DNNs, the training problems are often nonconvex and hence the assumption may not always hold. Surprisingly, overparameterized models can lead to tractable training problems. Indeed, (Allen-Zhu et al., 2019; Du et al., 2019; Zou et al., 2020; Nguyen & Mondelli, 2020; Nguyen, 2021) show that GD and SGD can find global solutions of the training problems for popular DNN families. For $T$ iterations, GD and SGD can find a solution with error $O(T^{-0.5})$. Those results suggests that $T = O(m)$ iterations are sufficient to ensure our assumption about $\epsilon_o(m)$. Allen-Zhu et al. (2019) show that $T = O(\log m)$ iterations are sufficient to ensure $\epsilon_o(m) = O(m^{-1})$.

Combining Theorems 3 with Lemma 1 will lead to the following for Dropout DNNs. Similar results can be shown for SN-DNNs.

**Theorem 5** (Dropout family). *Given the assumptions in Theorem 3, consider any $h_o \in \mathcal{H}_{dr}$. Let $h^* = \arg\min_{h \in \mathcal{H}_{dr}} F(P_x, h)$, and $\epsilon_o = F(\widehat{P}_x, h_o) - \min_{h \in \mathcal{H}_{dr}} F(\widehat{P}_x, h)$ be the optimization error of $h_o$ on a sample $\boldsymbol{S}$ of size $m$. For any constant $\delta \in (0, 1]$, with probability at least $1 - \delta$: $|F(P_x, h_o) - F(P_x, h^*)| \leq \epsilon_o + 2\left(C_{dr} L_f B + C\sqrt{\log 4 - 2\log \delta}\right) m^{-0.5}$*

**Corollary 2** (Consistency of Dropout DNNs). *Given the assumptions in Theorem 5, consider a learning algorithm $\mathcal{A}$ and family $\mathcal{H}_{dr}$. If, for any given sample $\boldsymbol{S}$ of size $m$, the learned function $h_o = \mathcal{A}(\mathcal{H}_{dr}, \boldsymbol{S})$ has optimization error at most $\epsilon_o(m)$ which is a decreasing function of $m$, then $\mathcal{A}$ is consistent with rate $\epsilon_o(m) + 2\left(C_{dr} L_f B + C\sqrt{\log 4 - 2\log \delta}\right) m^{-0.5}$.*

*Connection to overparameterization:* Contrary to classical wisdoms about overfitting, modern machine learning exhibits a strange phenonmenon: very rich models such as neural networks are trained to *exactly fit* (i.e., interpolate and $\epsilon_o = 0$) the data, but often obtain high accuracy on test data (Belkin et al., 2019; Zhang et al., 2021). Those models often belong to overparameterization regime where the number of parameters in a model is far larger than $m$. Such a strikingly strange behavior could not be explained by traditional learning theories (Zhang et al., 2021). Some works try to understand overparameterization in linear regression (Bartlett et al., 2020) and kernel regression (Liang et al., 2020). Some recent results (Kuzborskij & Szepesvári, 2021; Ji et al., 2021; Hu et al., 2021; Jacot et al., 2018) on consistency hold only for shallow neural networks with no more than 3 layers. However, consistency of deep neural networks remains largely open.

For overparameterized NNs with a suitable width, $T = O(m)$ iterations are sufficient for GD and SGD to achieve optimization error $\epsilon_o(m) = O(m^{-0.5})$ as discussed before. Combining this observation with Corollary 2 will reveal consistency with rate $O(m^{-0.5})$ for Dropout DNNs and SN-DNNs. To the best of our knowledge, this is the first consistency result for overparameterized DNNs which are truly deep and avoid the curse of dimensionality.

**Why are small consistency rates for high-capacity families sufficient to guarantee high generalization?** To see why, consider $Gap_B(h_o, \eta) = F(P_x, h_o) - F(P_x, \eta)$ which is the *Bayes gap* of an $h_o = \mathcal{A}(\mathcal{H}, \boldsymbol{S})$, where $\eta$ denotes the (unknown) true function we are trying to learn. Note that $Gap_B(h_o, \eta) = Cons(h_o, m) + F(P_x, h^*) - F(P_x, \eta)$, where $Cons(h_o, m) = F(P_x, h_o) - F(P_x, h^*)$ denotes the consistency rate. This decomposition suggests that a requirement of both $Cons(h_o, m)$ and $F(P_x, h^*)$ to be small will ensure a small $Gap_B(h_o, \eta)$, since $F(P_x, \eta)$ is independent of $\mathcal{H}$. In other words, a small consistency rate for high-capacity $\mathcal{H}$ is sufficient to guarantee high performance of $h_o$ on test data.

Overparameterized NNs often have a very high capacity. Some regularization methods can help us localize a subset $\mathcal{H}_g$ of the chosen NN family so that $\mathcal{H}_g$ has a small generalization gap. For example, in Theorem 3, we originally need to work with family $\mathcal{H} = \{h_{\mathcal{W}} : \|W_i\|_F \leq b_i\}$, but Dropout localizes a subset $\mathcal{H}_{dr} \subset \mathcal{H}$ having a small generalization gap. One should ensure that $\mathcal{H}_g$ still has a high capacity to produce a small optimization error. Interestingly, a small (even zero) optimization error is frequently observed in practice for overparameterized NNs (Zhang et al., 2021). In those cases, we can achieve a small consistency rate as shown in Corollary 2. Our work shows this property for Dropout and SN. Combining these arguments with the above sufficient condition will provide an answer for why those overparameterized NNs can work well on test data.

## 4 Generalization of GANs

This section presents a comprehensive analysis on generalization of GANs. We then discuss why Lipschitz constraint succeeds in practice.

*Notations:* Let $\boldsymbol{S} = \{x_1, ..., x_m, z_1, ..., z_m\}$ consist of $m$ i.i.d. samples from real distribution $P_d$ defined on a compact set $\mathcal{Z}_x \subset \mathbb{R}^{n_x}$ and $m$ i.i.d. samples from noise distribution $P_z$ defined on a compact set $\mathcal{Z}_z \subset \mathbb{R}^n$, $\widehat{P}_x$ and $\widehat{P}_z$ be the empirical distributions defined from $\boldsymbol{S}$ respectively. Denote $\mathcal{D}$ as the discriminator family and $\mathcal{G}$ as the generator family. Let $v(D, G, x, z) = \psi_1(D(x)) + \psi_2(1 - D(G(z)))$ be the loss defined from a real example $x \sim P_d$, a noise $z \sim P_z$, a discriminator $D \in \mathcal{D}$, and a generator $G \in \mathcal{G}$. Different choices of the *measuring functions* $(\psi_1, \psi_2)$ will lead to different GANs. For example, saturating GAN (Goodfellow et al., 2014) uses $\psi_1(x) = \psi_2(x) = \log(x)$; WGAN (Arjovsky et al., 2017) uses $\psi_1(x) = \psi_2(x) = x$; LSGAN (Mao et al., 2017; 2019) uses $\psi_1(x) = -(x + a)^2, \psi_2(x) = -(x + b)^2$ for some constants $a, b$; EBGAN (Zhao et al., 2017) uses $\psi_1(x) = x, \psi_2(x) = \max(0, r - x)$ for some constant $r$. We will often work with: $V(P_d, P_z, D, G) = \mathbb{E}_{x \sim P_d} \psi_1(D(x)) + \mathbb{E}_{z \sim P_z} \psi_2(1 - D(G(z))); V(P_d, \widehat{P}_z, D, G) = \mathbb{E}_{x \sim P_d} \psi_1(D(x)) + \mathbb{E}_{z \sim \widehat{P}_z} \psi_2(1 - D(G(z)))$ $V(\widehat{P}_d, P_z, D, G) = \mathbb{E}_{x \sim \widehat{P}_d} \psi_1(D(x)) + \mathbb{E}_{z \sim P_z} \psi_2(1 - D(G(z))); V(\widehat{P}_d, \widehat{P}_z, D, G) = \mathbb{E}_{x \sim \widehat{P}_d} \psi_1(D(x)) + \mathbb{E}_{z \sim \widehat{P}_z} \psi_2(1 - D(G(z)))$

In practice, we only have a finite sample $\boldsymbol{S}$ and an optimizer will solve $\min_{G \in \mathcal{G}} \max_{D \in \mathcal{D}} V(\widehat{P}_d, \widehat{P}_z, D, G)$ and return an approximate solution $(D_o, G_o)$, which can be different from the *training optimum* $(D_o^*, G_o^*)$ and *Nash solution* $(D^*, G^*)$, where

$$(D_o^*, G_o^*) = \arg\min_{G \in \mathcal{G}} \max_{D \in \mathcal{D}} V(\widehat{P}_d, \widehat{P}_z, D, G), \quad (D^*, G^*) = \arg\min_{G \in \mathcal{G}} \max_{D \in \mathcal{D}} V(P_d, P_z, D, G) \quad (2)$$

In learning theory, we often estimate $(V(P_d, P_z, D_o, G_o) - V(\widehat{P}_d, \widehat{P}_z, D_o, G_o))$ to see generalization. However a small bound on this quantity may not be enough, since $V(P_d, P_z, D_o, G_o)$ can be far from the best $V(P_d, P_z, D^*, G^*)$. Another way (Bousquet et al., 2004) is to see *How good is $(D_o, G_o)$ compared to the Nash solution $(D^*, G^*)$?* In other words, we basically need to estimate the difference $|V(P_d, P_z, D_o, G_o) - V(P_d, P_z, D^*, G^*)| = |V(P_d, P_z, D_o, G_o) - \min_{G \in \mathcal{G}} \max_{D \in \mathcal{D}} V(P_d, P_z, D, G)|$ where $V(P_d, P_z, D_o, G_o)$ shows the quality of the fake distribution induced by generator $G_o$.

We first make the following error decomposition:

$$V(P_d, P_z, D_o, G_o) - V(P_d, P_z, D^*, G^*) = [V(P_d, P_z, D_o, G_o) - V(\widehat{P}_d, \widehat{P}_z, D_o, G_o)] +$$

$$[V(\widehat{P}_d, \widehat{P}_z, D_o, G_o) - V(\widehat{P}_d, \widehat{P}_z, D_o^*, G_o^*)] + [V(\widehat{P}_d, \widehat{P}_z, D_o^*, G_o^*) - V(P_d, P_z, D^*, G^*)] \quad (3)$$

The first term $(V(P_d, P_z, D_o, G_o) - V(\widehat{P}_d, \widehat{P}_z, D_o, G_o))$ in the right-hand side of (3) shows the difference between the population and empirical losses of a specific solution $(D_o, G_o)$. The second term $(V(\widehat{P}_d, \widehat{P}_z, D_o, G_o) - V(\widehat{P}_d, \widehat{P}_z, D_o^*, G_o^*))$ is in fact the *Optimization error* incurred by the optimizer. This error depends strongly on the capacity of the chosen optimizer. The third term $(V(\widehat{P}_d, \widehat{P}_z, D_o^*, G_o^*) - V(P_d, P_z, D^*, G^*))$ is optimizer-independent and strongly depends on the capacity of both families $(\mathcal{D}, \mathcal{G})$, since both $V(\widehat{P}_d, \widehat{P}_z, D_o^*, G_o^*)$ and $V(P_d, P_z, D^*, G^*)$ are optimizer-independent. We call this term *Joint error* of $(\mathcal{D}, \mathcal{G})$. In the next subsections, we will provide upper bounds on both the error of $(D_o, G_o)$ and joint error of $(\mathcal{D}, \mathcal{G})$, and then generalization bounds that take the optimization error into account.

In the later discussions, we will often use the following assumptions and notation $L = L_\psi L_d L_g$ which upper bounds the Lipschitz constant of the loss $v(D, G, x, z)$.

**Assumption 1.** *$\psi_1$ and $\psi_2$ are $L_\psi$-Lipschitz continuous w.r.t. their inputs on a compact domain and upper-bounded by constant $C \geq 0$.*

**Assumption 2.** *Each generator $G \in \mathcal{G}$ is $L_g$-Lipschitz continuous w.r.t its input $z$ over a compact set $\mathcal{Z}_z \subset \mathbb{R}^n$ with diameter at most $B_z$.*

**Assumption 3.** *Each discriminator $D \in \mathcal{D}$ is $L_d$-Lipschitz continuous w.r.t its input $x$ over a compact set $\mathcal{Z}_x \subset \mathbb{R}^{n_x}$ with diameter at most $B_x$.*

These assumptions are reasonable and satisfied by various GANs. For example, WGAN, LSGAN, EBGAN naturally satisfy Assumption 1, while saturating GAN will satisfy it if we constraint the output of $D$ to be in $[\alpha, \beta] \subset (0, 1)$ as often used in practice. Spectral normalization and gradient penalty are popular techniques to regularize $D$ and are crucial for large-scale generators (Zhang et al., 2019; Karras et al., 2020b). Therefore Assumptions 3 and 2 are natural.

## 4.1 ERROR BOUNDS

The following result readily comes from Theorem 1.

**Corollary 3.** *Given the assumptions (1, 2, 3), for any $\delta \in (0, 1]$, $\lambda \in (0, B_z]$, with probability at least $1 - \delta$, we have*
$$\sup_{D \in \mathcal{D}, G \in \mathcal{G}} |V(P_d, P_z, D, G) - V(P_d, \widehat{P}_z, D, G)| \leq L\lambda + \frac{C}{\sqrt{m}} \sqrt{\lceil B_z^n \lambda^{-n} \rceil \log 4 - 2 \log \delta}$$

This corollary tells the generalization of any generator $G \in \mathcal{G}$, and can be further tighten by using Theorem 3 when using SN or Dropout. To see generalization of both players $(D_o, G_o)$, observe that $|V(P_d, P_z, D_o, G_o) - V(\widehat{P}_d, \widehat{P}_z, D_o, G_o)| \leq \sup_{D \in \mathcal{D}, G \in \mathcal{G}} |V(\widehat{P}_d, \widehat{P}_z, D, G) - V(P_d, P_z, D, G)|$. The following theorem provides an upper bound whose proof appears in Appendix B.

**Theorem 6.** *Denote $\epsilon(\mathcal{D}, \mathcal{G}) = \sup_{D \in \mathcal{D}, G \in \mathcal{G}} |V(\widehat{P}_d, \widehat{P}_z, D, G) - V(P_d, P_z, D, G)|$. Given the assumptions (1, 2, 3), for any constants $\delta, \delta_x \in (0, 1]$,*

***(General family)*** *for any $\lambda \in (0, B_z]$, $\lambda_x \in (0, B_x]$, with probability at least $1 - \delta - \delta_x$:*
$$\epsilon(\mathcal{D}, \mathcal{G}) \leq L\lambda + \frac{C}{\sqrt{m}} \sqrt{\lceil B_z^n \lambda^{-n} \rceil \log 4 - 2 \log \delta} + L_\psi L_d \lambda_x + \frac{C}{\sqrt{m}} \sqrt{\lceil B_x^{n_x} \lambda_x^{-n_x} \rceil \log 4 - 2 \log \delta_x}$$

***(D with spectral norm)*** *given the assumptions in Theorem 3, with probability at least $1 - 2\delta$:*
$$\epsilon(\mathcal{H}_{sn}, \mathcal{G}) \leq [C_{sn} L_\psi L_g B_z + 2C\sqrt{\log 4 - 2 \log \delta} + C_{sn} L_\psi B_x] m^{-0.5}$$

*(D **with Dropout**) given the assumptions in Theorem 3, with probability at least $1 - 2\delta$:*
$$\epsilon(\mathcal{H}_{dr}, \mathcal{G}) \leq [C_{dr}L_\psi L_g B_z + 2C\sqrt{\log 4 - 2\log \delta} + C_{dr}L_\psi B_x]m^{-0.5}$$

For many models, such as WGAN, the measuring functions and $D$ are Lipschitz continuous w.r.t their inputs. Note that the generator in WGAN, LSGAN, and EBGAN will be Lipschitz continuous w.r.t $z$, if we use some regularization methods such as gradient penalty or spectral normalization for both players. Theorem 6 also suggests that **penalizing the zero-order** ($C$) **and first-order** ($L$) **informations of the loss can improve the generalization**. This provides a significant evidence for the important role of gradient penalty or spectral normalization for the success of some large-scale generators (Zhang et al., 2019; Brock et al., 2019; Karras et al., 2020b).

It is worth observing that a small Lipschitz constant of the loss not only requires that both discriminator and generator are Lipschitz continuous w.r.t their inputs, but also requires Lipschitz continuity of the loss w.r.t both players. Most existing efforts focus on the players in GANs, and leave the loss open. Constraining on either $D$ or $G$ may be insufficient to ensure Lipschitz continuity of the loss.

One advantage of the generalization bounds in Theorem 6 is that the upper bounds on $|V(\widehat{P}_d, \widehat{P}_z, D, G) - V(P_d, P_z, D, G)|$ hold true for any particular $(D, G)$ in their families. Meanwhile, the existing generalization bounds (Arora et al., 2017; Zhang et al., 2018; Jiang et al., 2019; Wu et al., 2019; Husain et al., 2019) hold true conditioned on the best discriminator. Hence the bounds in Theorem 6 are more practical than existing ones, since $D$ is not trained to optimality before training $G$ in practical implementations of GANs.

Next we consider the joint error $V(\widehat{P}_d, \widehat{P}_z, D_o^*, G_o^*) - V(P_d, P_z, D^*, G^*)$ of both families $(\mathcal{D}, \mathcal{G})$. Such a quantity also shows the goodness of the training optimum $(D_o^*, G_o^*)$ compared with the Nash solution $(D^*, G^*)$. It is worth observing that $|V(\widehat{P}_d, \widehat{P}_z, D_o^*, G_o^*) - V(P_d, P_z, D^*, G^*)| = |\min_{G \in \mathcal{G}} \max_{D \in \mathcal{D}} V(\widehat{P}_d, \widehat{P}_z, D, G) - \min_{G \in \mathcal{G}} \max_{D \in \mathcal{D}} V(P_d, P_z, D, G)|$ measures the quality of the best players given a finite number of samples only, and such error does not depend on any optimizer. Hence it represents the *Joint capacity* of both generator and discriminator families. The following theorem provides an upper bound whose proof appears in Appendix B.

**Theorem 7.** *Given the assumptions (1, 2, 3), for any constants $\delta, \delta_x \in (0, 1]$, $\lambda \in (0, B_z]$, $\lambda_x \in (0, B_x]$, with probability at least $1 - \delta - \delta_x$: $|V(\widehat{P}_d, \widehat{P}_z, D_o^*, G_o^*) - V(P_d, P_z, D^*, G^*)| \leq L\lambda + \frac{C}{\sqrt{m}}\sqrt{\lceil B_z^n \lambda^{-n} \rceil \log 4 - 2\log \delta} + L_\psi L_d \lambda_x + \frac{C}{\sqrt{m}}\sqrt{\lceil B_x^{n_x} \lambda_x^{-n_x} \rceil \log 4 - 2\log \delta_x}.$*

This bound on joint capacity of $(\mathcal{D}, \mathcal{G})$ is loose, since few informations about those families are used. We can tighten this bound when using SN or Dropout for $\mathcal{D}$, similar with Theorem 6.

## 4.2 FROM OPTIMIZATION ERROR TO GENERALIZATION

Finally we make a bidge between optimization error and generalization. The decomposition (3) contains three components, for which the first component is bounded in Theorem 6 while the third component is bounded in Theorem 7. Combining those observations will lead to the following result.

**Theorem 8** (Generalization bounds for GANs). *Assume the assumptions (1, 2, 3) and the optimization error $|V(\widehat{P}_d, \widehat{P}_z, D_o, G_o) - \min_{G \in \mathcal{G}} \max_{D \in \mathcal{D}} V(\widehat{P}_d, \widehat{P}_z, D, G)| \leq \epsilon_o$. Denote $\epsilon_{cons}(\mathcal{D}, \mathcal{G}) = |V(P_d, P_z, D_o, G_o) - V(P_d, P_z, D^*, G^*)|$. For any constants $\delta, \delta_x \in (0, 1]$,*

1. *(**General family**) for any $\lambda \in (0, B_z]$, $\lambda_x \in (0, B_x]$, with probability at least $1 - \delta - \delta_x$: $\epsilon_{cons}(\mathcal{D}, \mathcal{G}) \leq \epsilon_o + 2L\lambda + \frac{2C}{\sqrt{m}}\sqrt{\lceil B_z^n \lambda^{-n} \rceil \log 4 - 2\log \delta} + 2L_\psi L_d \lambda_x + \frac{2C}{\sqrt{m}}\sqrt{\lceil B_x^{n_x} \lambda_x^{-n_x} \rceil \log 4 - 2\log \delta_x}.$*

2. *(**Spectral norm**) given the assumptions in Theorem 3, $\mathcal{D} \equiv \mathcal{H}_{sn}$, with probability at least $1 - 2\delta$: $\epsilon_{cons}(\mathcal{H}_{sn}, \mathcal{G}) \leq \epsilon_o + 2[C_{sn}L_\psi L_g B_z + 2C\sqrt{\log 4 - 2\log \delta} + C_{sn}L_\psi B_x]m^{-0.5}$*

3. *(**Dropout**) given the assumptions in Theorem 3, $\mathcal{D} \equiv \mathcal{H}_{dr}$, with probability at least $1 - 2\delta$: $\epsilon_{cons}(\mathcal{H}_{dr}, \mathcal{G}) \leq \epsilon_o + 2[C_{dr}L_\psi L_g B_z + 2C\sqrt{\log 4 - 2\log \delta} + C_{dr}L_\psi B_x]m^{-0.5}$*

Theorems 6 and 8 provide us a comprehensive view about generalization of GANs. Note that their assumptions are naturally met in practice as pointed out before. For the first time in the GAN litera-

ture, our work reveals that GANs can avoid the curse of dimensionality when choosing appropriate $(\mathcal{D}, \mathcal{G})$. Furthermore, a logarithmic (in $m$) number of layers are sufficient for each player. Although this work shows this property for DNNs with spectral norm or Dropout. We believe that this property can hold for many other DNN families.

One important implication from Theorem 8 is that GANs can be consistent under suitable conditions. An example condition is overparameterization, for which the optimization error can be zero. Our experiments in Appendix F provide a good evidence for this conjecture as the well-trained discriminators often reach Nash equilibria. A recent investigation about optimization of overparameterized GANs appears in (Balaji et al., 2021). We leave this door open for the readers.

### 4.3 WHY A LIPSCHITZ CONSTRAINT IS CRUCIAL

Various works (Guo et al., 2019; Jenni & Favaro, 2019; Qi, 2020; Arjovsky et al., 2017; Gulrajani et al., 2017; Roth et al., 2017; Miyato et al., 2018; Zhou et al., 2019; Thanh-Tung et al., 2019; Jiang et al., 2019; Tanielian et al., 2020; Xu et al., 2020) try to ensure Lipschitz continuity of the discriminator or generator. The most popular techniques are gradient penalty (Gulrajani et al., 2017) and spectral normalization (SN) (Miyato et al., 2018). Those techniques are really useful for different losses (Fedus et al., 2018) and high-capacity architectures (Kurach et al., 2019). From a large-scale evaluation, Kurach et al. (2019) found that gradient penalty can help the performance of GANs but does not stabilize the training, whereas using SN for $G$ only is insufficient to ensure stability (Brock et al., 2019). Some recent large-scale generators (Brock et al., 2019; Zhang et al., 2019; Karras et al., 2020b) use gradient penalty or SN to ensure their successes. Data augmentation (Zhao et al., 2020a;b; Tran et al., 2021; Karras et al., 2020a) also contributes to the excellent performance of GANs in practice, due to implicitly penalizing the Lipschitz constant of the loss (see Appendix D for explanation). Those empirical observations without a theory poses a long mystery of *why can imposing a Lipschitz constraint help GANs to perform well?* This work provides an answer:

▷ Theorems 6 and 8 show that a Lipschitz constraint on one player ($D$ or $G$) can help, but may be not enough. A penalty on the first-order ($L$) information of the loss can lead to better generalization.

▷ Spectral normalization (Miyato et al., 2018) is a popular technique to regularize GANs. Using SN, the spectral norms of the weight matrices are often small in practice, and hence the Lipschitz constant of $D$ (or $G$) can be exponentially small when using SN. In those cases, the assumptions of Theorem 8 are satisfied. Therefore the generalization bound in Theorem 8 is tight and supports well the success of spectrally-normalized GANs (Miyato et al., 2018; Zhang et al., 2019).

▷ Dropout and SN are really efficient to control the complexity of the players and provide tight generalization bounds.

▷ WGAN (Arjovsky et al., 2017) naturally requires $D$ to be 1-Lipschitz continuous. Weight clipping is used so that every element of network weights belongs to $[-c, c]$ for some constant $c$. For some choices, e.g. $c = 0.01$ in (Arjovsky et al., 2017), the spectral norm of the weight matrix at each layer can be smaller than 1.[2] In those cases the Lipschitz constant of $D$ can be exponentially small, leading to tight bounds in Theorem 8 and better generalization.

▷ SN, gradient penalty, and data augmentation are crucial parts of large-scale GANs (Brock et al., 2019; Zhang et al., 2019; Karras et al., 2020b). As a result, Theorems 6 and 8 provide a strong support for their success in practice.

▷ Our experiments with SN in Appendix F indeed show that SN can reduce the Lipschitz constants of the players and the loss. However, when SN is overused, the trained players can get underfitting and may hurt generalization. A reason is that an underfitted model can have a bad population loss and high optimization error.

## 5 CONCLUSION

We have presented a simple way to analyze generalization of various complex models that are hard for traditional learning theories. Some successful applications were done and made a significant step toward understanding DNNs and GANs. One limitation of our bounds is that the optimization aspect is left open.

---

[2]For $c = 0.01$, if the number of units at each layer is no more than 100, then the Frobenius norm of the weight matrice at each layer is smaller than 1, and so is for the spectral norm.

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

# A  LIPSCHITZ CONTINUITY ⇒ GENERALIZATION

This section provides the proofs for the theorems of Section 3. Let $\mathcal{X} = \bigcup_{i=1}^{N} \mathcal{X}_i$ be a partition of $\mathcal{X}$ into $N$ disjoint subsets. We use the following definition from Xu & Mannor (2012).

**Definition 2** (Robustness). *An algorithm $\mathcal{A}$ is $(N, \epsilon)$-**robust**, for $\epsilon : \mathcal{X}^m \to \mathbb{R}$, if the following holds for all $\boldsymbol{S} \in \mathcal{X}^m$:*
$\forall s \in \boldsymbol{S}, \forall x \in \mathcal{X}, \forall i \in \{1, ..., N\}$, *if $s, x \in \mathcal{X}_i$ then $|f(\mathcal{A}(\mathcal{H}, \boldsymbol{S}), s) - f(\mathcal{A}(\mathcal{H}, \boldsymbol{S}), x)| \leq \epsilon(\boldsymbol{S})$.*

Basically, a robust algorithm will learn a hypothesis which ensures that the losses of two similar data instances should be the same. A small change in the input leads to a small change in the loss of the learnt hypothesis. In other words, the robustness ensures that each testing sample which is close to the training dataset will have a similar loss with that of the closest training samples. Therefore, the hypothesis $\mathcal{A}(\mathcal{H}, \boldsymbol{S})$ will generalize well over the areas around $\boldsymbol{S}$.

**Theorem 9** (Xu & Mannor (2012)). *If a learning algorithm $\mathcal{A}$ is $(N, \epsilon)$-robust and the training data $\boldsymbol{S}$ is an i.i.d. sample from distribution $P_x$, then for any $\delta \in (0, 1]$ we have the following with probability at least $1 - \delta$: $|F(P_x, \mathcal{A}(\mathcal{H}, \boldsymbol{S})) - F(\widehat{P}_x, \mathcal{A}(\mathcal{H}, \boldsymbol{S}))| \leq \epsilon(\boldsymbol{S}) + C\sqrt{(N \log 4 - 2 \log \delta)/m}.$*

This theorem formally makes the important connection between robustness of an algorithm and generalization. If an algorithm is robust, then its resulting hypotheses can generalize. One important implication of this result is that we should ensure the robustness of a learning algorithm. However, it is nontrivial to do so in general.

Let us have a closer look at robustness. $\epsilon(\boldsymbol{S})$ in fact bounds the amount of change in the loss with respect to a change in the input given a fixed hypothesis. This observation suggests that robustness closely resembles the concept of Lipschitz continuity. Remember that a function $y : \mathcal{X} \to \mathbb{Y}$ is said to be *L-Lipschitz continuous* if $d_y(y(x), y(x')) \leq L d_x(x, x')$ for any $x, x' \in \mathcal{X}$, where $d_x$ is a metric on $\mathcal{X}$, $d_y$ is a metric on $\mathbb{Y}$, and $L \geq 0$ is the Lipschitz constant. Therefore, we establish the following connection between robustness and Lipschitz continuity.

**Lemma 2.** *Given any constant $\lambda > 0$, consider a loss $f : \mathcal{H} \times \mathcal{X} \to \mathbb{R}$, where $\mathcal{X} \subset \mathbb{R}^{n_x}$ is compact, $B = diam(\mathcal{X}) = \max_{x, x' \in \mathcal{X}} \|x - x'\|_\infty$, $N = \lceil B^{n_x} \lambda^{-n_x} \rceil$. If for any $h \in \mathcal{H}$, $f(h, x)$ is L-Lipschitz continuous w.r.t input $x$, then any algorithm $\mathcal{A}$ that maps $\mathcal{X}^m$ to $\mathcal{H}$ is $(N, L\lambda)$-robust.*

**Proof:** It is easy to see that there exist $N = \lceil (B/\lambda)^{n_x} \rceil$ disjoint $n_x$-dimensional cubes, each with edge length of $\lambda$, satisfying that their union covers $\mathcal{X}$ completely since $\mathcal{X}$ is compact. Let $\mathbb{C}_k$ be one of those cubes, indexed by $k$, and $\mathcal{X}_k = \mathcal{X} \cap \mathbb{C}_k$. We can write $\mathcal{X} = \bigcup_{k=1}^{N} \mathcal{X}_k$.

Consider any $s, x \in \mathcal{X}$. If both $s$ and $x$ belong to the same $\mathcal{X}_k$ for some $k$, then we have $\left| f(\mathcal{A}(\mathcal{H}, \boldsymbol{S}), s) - f(\mathcal{A}(\mathcal{H}, \boldsymbol{S}), x) \right| \leq L \|s - x\|_\infty \leq L\lambda$ for any algorithm $\mathcal{A}$ and any $\boldsymbol{S} \in \mathcal{X}^m$ due to the Lipschitz continuity of $f$, completing the proof.

$\square$

**Proof of Theorem 1:** For any $h \in \mathcal{H}$ and dataset $\boldsymbol{S}$, there exists an algorithm $\mathcal{A}$ that maps $\boldsymbol{S}$ to $h$, i.e., $h = \mathcal{A}(\mathcal{H}, \boldsymbol{S})$. Lemma 2 tells that $\mathcal{A}$ is $(\lceil B^{n_x} \lambda^{-n_x} \rceil, L\lambda)$-robust for any $\lambda \in (0, B]$. Theorem 9 implies $|F(P_x, h) - F(\widehat{P}_x, h)| \leq L\lambda + \frac{C}{\sqrt{m}} \sqrt{\lceil B^{n_x} \lambda^{-n_x} \rceil \log 4 - 2 \log \delta}$ with probability at least $1 - \delta$, for any constants $\delta \in (0, 1]$ and $\lambda \in (0, B]$. Since this bound holds true for any $h \in \mathcal{H}$, we conclude

$$\sup_{h \in \mathcal{H}} |F(P_x, h) - F(\widehat{P}_x, h)| \leq L\lambda + \frac{C}{\sqrt{m}} \sqrt{\lceil B^{n_x} \lambda^{-n_x} \rceil \log 4 - 2 \log \delta}$$

The second statement is an application of the first one by taking $\lambda = Bm^{-\alpha/n_x}$ and $\delta = 2\exp(-0.5m^\alpha)$. Indeed,

$$
\begin{aligned}
\sup_{h \in \mathcal{H}} |F(P_x, h) - F(\widehat{P}_x, h)| &\leq L\lambda + \frac{C}{\sqrt{m}} \sqrt{\lceil B^{n_x} \lambda^{-n_x} \rceil \log 4 - 2\log \delta} \\
&\leq LBm^{-\frac{\alpha}{n_x}} + \frac{C}{\sqrt{m}} \sqrt{\lceil m^\alpha \rceil \log 4 - \log 4 + m^\alpha} \\
&\leq LBm^{-\frac{\alpha}{n_x}} + \frac{C}{\sqrt{m}} \sqrt{m^\alpha \log 4 + m^\alpha} \\
&\leq LBm^{-\frac{\alpha}{n_x}} + Cm^{-\frac{\alpha}{n_x}} \sqrt{(1 + \log 4)m^{-1 + \frac{n_x+2}{n_x}\alpha}} \\
&\leq (LB + 2C)m^{-\frac{\alpha}{n_x}}
\end{aligned}
$$

The last inequality holds because $-1 + \frac{n_x+2}{n_x}\alpha \leq 0$ and hence $m^{-1+\frac{n_x+2}{n_x}\alpha} \leq 1$, completing the proof.

$\square$

## A.1 DEEP NEURAL NETWORKS THAT AVOID THE CURSE OF DIMENSIONALITY

**Proof of Theorem 2:** Denote $h_0(x) = x, h_1(x) = \sigma_1(W_1 h_0(x)), ..., h_K(x) = \sigma_K(W_K h_{K-1}(x))$. By definition, the Lipschitz constant of a function $g(h)$ is defined to be $\|g\|_{Lip} = \sup_{\|h\|_2 \leq 1} \sigma(\nabla g(h))$, where $\sigma(B)$ is the spectral norm of matrix $B$. For a linear function we have $\|Wh\|_{Lip} = \sup_{\|h\|_2 \leq 1} \sigma(Wh) = \|W\|_\sigma$. Since $\sigma_i$ is $\rho_i$-Lipschitz for any $i$, we have

$$
\begin{aligned}
\|h_K(x)\|_{Lip} &\leq \rho_K \|W_K h_{K-1}(x)\|_{Lip} && (4) \\
&\leq \rho_K \|W_K\|_\sigma \|h_{K-1}(x)\|_{Lip} && (5) \\
&\leq \rho_K \rho_{K-1} \|W_K\|_\sigma \|W_{K-1}\|_\sigma \|h_{K-2}(x)\|_{Lip} && (6) \\
&... && (7) \\
&\leq \prod_{k=1}^{K} \rho_k \|W_k\|_\sigma && (8) \\
&\leq \prod_{k=1}^{K} \rho_k s_k && (9)
\end{aligned}
$$

which proves the first statement.

Next consider a neural network $h_{\mathcal{A}}$ trained with Dropout (Srivastava et al., 2014). At each minibatch $t$ of the training phase, we randomly sample a thin sub-network of $h_{\mathcal{W}}$, compute the gradients $g^{(t)}(x)$ given the minibatch data, and then update each weight matrice as

$$
W_i^{(t)} := Normalize(\hat{W}_i^{(t)})_c \text{ where } \hat{W}_i^{(t)} := W_i^{(t-1)} - \eta g_i^{(t)}(x) \qquad (10)
$$

where $Normalize(\hat{W}_i^{(t)})_c$ is the normalization so that $\|W_i^{(t)}\|_F \leq c_i \leq b_i$ for some tuning constant $c_i$ and any $i$, and $\eta$ is the learning rate.

After training (with $T$ minibatchs), the network weights are scaled as $W_i^{(T)} := qW_i^{(T)}$ for any $i$, where $q$ is the drop rate. This implies that after training, we obtain a neural network $h_{\mathcal{W},q}$ with all weight matrices satisfying $\|W_i\|_F \leq qb_i$. By using the same arguments as above, we have

$$\|h_{\mathcal{W},q}(x)\|_{Lip} \quad \leq \quad \prod_{k=1}^{K} \rho_k \|W_k\|_\sigma \tag{11}$$

$$\leq \quad \prod_{k=1}^{K} \rho_k \|W_k\|_F \tag{12}$$

$$\leq \quad \prod_{k=1}^{K} \rho_k q b_k \tag{13}$$

where we have used the fact that $\|B\|_\sigma \leq \|B\|_F$ for any $B$, completing the proof.

□

**Proof of Theorem 3:** Let $L$ be the Lipschitz constant of loss $f(h, x)$ w.r.t $x$. The basic property of Lipschitz functions and composition shows that $L \leq L_f L_h$. For any $h \in \mathcal{H}_{dr}$, we have $L_h \leq q^K C_{dr}$ owing to Theorem 2. Hence $L \leq L_f q^K C_{dr} \leq L_f C_{dr} m^{-0.5}$.

Taking $\lambda = B$, Theorem 1 tells that

$$\sup_{h \in \mathcal{H}_{dr}} |F(P_x, h) - F(\widehat{P}_x, h)| \leq LB + C\sqrt{(\log 4 - 2\log \delta)/m} \leq BL_f C_{dr} m^{-0.5} + C\sqrt{(\log 4 - 2\log \delta)/m}$$

Similar arguments can be used for family $\mathcal{H}_{sn}$, completing the proof.

□

## A.2 CONSISTENCY PROOF

**Proof of Lemma 1:** We have

$$|F(P_x, h_o) - F(P_x, h^*)|$$

$$= \quad |F(P_x, h_o) - F(\widehat{P}_x, h_o) + F(\widehat{P}_x, h_o) - \min_{h \in \mathcal{H}} F(\widehat{P}_x, h) + \min_{h \in \mathcal{H}} F(\widehat{P}_x, h) - \min_{h \in \mathcal{H}} F(P_x, h)|$$

$$\leq \quad |F(P_x, h_o) - F(\widehat{P}_x, h_o)| + |F(\widehat{P}_x, h_o) - \min_{h \in \mathcal{H}} F(\widehat{P}_x, h)| + |\min_{h \in \mathcal{H}} F(\widehat{P}_x, h) - \min_{h \in \mathcal{H}} F(P_x, h)| \tag{14}$$

$$\leq \quad |F(P_x, h_o) - F(\widehat{P}_x, h_o)| + \epsilon_o + \sup_{h \in \mathcal{H}} |F(\widehat{P}_x, h) - F(P_x, h)| \tag{15}$$

$$\leq \quad \epsilon_o + 2\sup_{h \in \mathcal{H}} |F(\widehat{P}_x, h) - F(P_x, h)|$$

where we have used Lemma 3 to derive (15) from (14).

□

## B PROOFS OF THE MAIN THEOREMS FOR GANS

**Proof of Corollary 3:** Observe that

$$\sup_{D \in \mathcal{D}, G \in \mathcal{G}} |V(P_d, P_z, D, G) - V(P_d, \widehat{P}_z, D, G)| = \sup_{D \in \mathcal{D}, G \in \mathcal{G}} |\mathbb{E}_{z \sim P_z} \psi_2(1 - D(G(z))) -$$

$\mathbb{E}_{z \sim \widehat{P}_z} \psi_2(1 - D(G(z)))|$. Since $\widehat{P}_z$ is an empirical version of $P_z$, applying Theorem 1 will provide the generalization bounds for $\sup_{D \in \mathcal{D}, G \in \mathcal{G}} |\mathbb{E}_{z \sim P_z} \psi_2(1 - D(G(z))) - \mathbb{E}_{z \sim \widehat{P}_z} \psi_2(1 - D(G(z)))|$.

The same arguments can be done for $\sup_{D \in \mathcal{D}, G \in \mathcal{G}} |V(\widehat{P}_d, P_z, D, G) - V(\widehat{P}_d, \widehat{P}_z, D, G)|$, completing the proof.

□

**Proof of Theorem 6:** Observe that
$$|V(\widehat{P}_d, \widehat{P}_z, D, G) - V(P_d, P_z, D, G)|$$

$$= \left| \mathbb{E}_{x \sim \widehat{P}_d} \psi_1(D(x)) + \mathbb{E}_{z \sim \widehat{P}_z} \psi_2(1 - D(G(z))) - \mathbb{E}_{x \sim P_d, z \sim P_z} v(D, G, x, z) \right| \quad (16)$$

$$\leq \quad |\mathbb{E}_{z \sim P_z} \psi_2(1 - D(G(z))) - \mathbb{E}_{z \sim \widehat{P}_z} \psi_2(1 - D(G(z)))| + |\mathbb{E}_{x \sim P_d} \psi_1(D(x)) - \mathbb{E}_{x \sim \widehat{P}_d} \psi_1(D(x))|$$

Therefore
$$\sup_{G \in \mathcal{G}, D \in \mathcal{D}} |V(\widehat{P}_d, \widehat{P}_z, D, G) - V(P_d, P_z, D, G)|$$

$$\leq \quad \sup_{G \in \mathcal{G}, D \in \mathcal{D}} |\mathbb{E}_{z \sim P_z} \psi_2(1 - D(G(z))) - \mathbb{E}_{z \sim \widehat{P}_z} \psi_2(1 - D(G(z)))|$$

$$+ \sup_{G \in \mathcal{G}, D \in \mathcal{D}} |\mathbb{E}_{x \sim P_d} \psi_1(D(x)) - \mathbb{E}_{x \sim \widehat{P}_d} \psi_1(D(x))| \quad (17)$$

Theorem 1 shows that $\sup_{G \in \mathcal{G}, D \in \mathcal{D}} |\mathbb{E}_{z \sim P_z} \psi_2(1 - D(G(z))) - \mathbb{E}_{z \sim \widehat{P}_z} \psi_2(1 - D(G(z)))| \leq L\lambda + C\sqrt{(\lceil B_z^n \lambda^{-n} \rceil \log 4 - 2 \log \delta)/m}$, with probability at least $1 - \delta$, for any constants $\delta \in (0, 1]$ and $\lambda \in (0, B_z]$. Similarly, we have $\sup_{G \in \mathcal{G}, D \in \mathcal{D}} |\mathbb{E}_{x \sim P_d} \psi_1(D(x)) - \mathbb{E}_{x \sim \widehat{P}_d} \psi_1(D(x))| \leq L_\psi L_d \lambda_x + C\sqrt{(\lceil B_x^{n_x} \lambda_x^{-n_x} \rceil \log 4 - 2 \log \delta_x)/m}$, with probability at least $1 - \delta_x$, for any constants $\delta_x \in (0, 1]$ and $\lambda_x \in (0, B_x]$. Combining these bounds with (17) and the union bound will lead to the first statement of the theorem.

For the second and third statements, we choose $\lambda = B_z, \lambda_x = B_x, \delta = \delta_x$. Using the bounds for the Lipschitz constant of $D$ in Theorem 3 will complete the proof.

□

**Proof of Theorem 7:** By definition, $(D_o^*, G_o^*) = \arg\min_{G \in \mathcal{G}} \max_{D \in \mathcal{D}} V(\widehat{P}_d, \widehat{P}_z, D, G)$ and $(D^*, G^*) = \arg\min_{G \in \mathcal{G}} \max_{D \in \mathcal{D}} V(P_d, P_z, D, G)$.

Therefore
$$|V(\widehat{P}_d, \widehat{P}_z, D_o^*, G_o^*) - V(P_d, P_z, D^*, G^*)|$$

$$= \quad |\min_{G \in \mathcal{G}} \max_{D \in \mathcal{D}} V(\widehat{P}_d, \widehat{P}_z, D, G) - \min_{G \in \mathcal{G}} \max_{D \in \mathcal{D}} V(P_d, P_z, D, G)| \quad (18)$$

$$\leq \quad \max_{G \in \mathcal{G}} |\max_{D \in \mathcal{D}} V(\widehat{P}_d, \widehat{P}_z, D, G) - \max_{D \in \mathcal{D}} V(P_d, P_z, D, G)| \quad (19)$$

$$\leq \quad \max_{G \in \mathcal{G}} \max_{D \in \mathcal{D}} |V(\widehat{P}_d, \widehat{P}_z, D, G) - V(P_d, P_z, D, G)| \quad (20)$$

$$\leq \quad L\lambda + C\sqrt{(\lceil B_z^n \lambda^{-n} \rceil \log 4 - 2 \log \delta)/m} + L_\psi L_d \lambda_x + C\sqrt{(\lceil B_x^{n_x} \lambda_x^{-n_x} \rceil \log 4 - 2 \log \delta_x)/m} \quad (21)$$

where we have used Lemma 3 to derive (20) from (19) and (21) from (20). The last inequality comes from Theorem 6, completing the proof.

□

**Lemma 3.** *Assume that $h_1$ and $h_2$ are continuous functions defined on a compact set $\mathcal{Z}_x$. Then*
$$|\max_{x \in \mathcal{Z}_x} h_1(x) - \max_{x \in \mathcal{Z}_x} h_2(x)| \leq \max_{x \in \mathcal{Z}_x} |h_1(x) - h_2(x)|$$
$$|\min_{x \in \mathcal{Z}_x} h_1(x) - \min_{x \in \mathcal{Z}_x} h_2(x)| \leq \max_{x \in \mathcal{Z}_x} |h_1(x) - h_2(x)|$$

*Proof:* Denote $x_1^* = \arg\max_{x \in \mathcal{Z}_x} h_1(x), x_2^* = \arg\max_{x \in \mathcal{Z}_x} h_2(x)$. It is easy to see that
$$h_1(x_2^*) - h_2(x_2^*) \leq h_1(x_1^*) - h_2(x_2^*) \leq h_1(x_1^*) - h_2(x_1^*)$$
Therefore
$$|\max_{x \in \mathcal{Z}_x} h_1(x) - \max_{x \in \mathcal{Z}_x} h_2(x)| = |h_1(x_1^*) - h_2(x_2^*)| \leq \max_{x \in \mathcal{Z}_x} |h_1(x) - h_2(x)|.$$

We can rewrite $|\min_{x \in \mathcal{Z}_x} h_1(x) - \min_{x \in \mathcal{Z}_x} h_2(x)| = |-\max_{x \in \mathcal{Z}_x}(-h_1(x)) + \max_{x \in \mathcal{Z}_x}(-h_2(x))| \leq \max_{x \in \mathcal{Z}_x} |h_1(x) - h_2(x)|$, completing the proof.

□

## C  GANs and Autoencoders

### C.1  Tightness of the bounds for GANs

Note that our bounds in Theorems 6 and 8 in general are not tight in terms of sample complexity and dimensionality. Taking $\lambda = B_z m^{-1/(n+2)}, \delta = 2\exp(-0.5m^{n/(n+2)}), \lambda_x = B_x m^{-1/(n_x+2)}, \delta_x = 2\exp(-0.5m^{n_x/(n_x+2)})$, Theorem 6 provides $\sup_{D\in\mathcal{D},G\in\mathcal{G}} |V(\widehat{P}_d,\widehat{P}_z,D,G) - V(P_d,P_z,D,G)| \leq O(m^{-1/(n+2)} + m^{-1/(n_x+2)})$. This bound $O(m^{-1/(n+2)} + m^{-1/(n_x+2)})$ surpasses the previous best bound $O(m^{-1/(1.5n)} + m^{-1/(1.5n_x)})$ in the GAN literature (Husain et al., 2019).

### C.2  Sample-efficient bounds for Autoencoders

Husain et al. (2019) did a great job at connecting GANs and Autoencoder models. They showed that the generator objective in $f$-GAN (Nowozin et al., 2016) is upper bounded by the objective of Wasserstein Autoencoders (WAE) (Tolstikhin et al., 2018). Under some suitable conditions, the two objectives equal. They further showed the bound:

$$|\max_{D\in\mathcal{D}} V(P_d,P_z,D,G) - \max_{D\in\mathcal{D}} V(\widehat{P}_d,\widehat{P}_z,D,G)| \leq O(m^{-1/s_d} + m^{-1/s_g}),$$

where $s_d > d^*(P_d)$ (the 1-upper Wasserstein dimension of $P_d$) and $s_g > d^*(P_g)$. We show in Appendix C.3 that $s_d > 1.5n_x, s_g > 1.5n$ even for a simple distribution, where $n_x$ is the dimensionality of real data, and $n$ is the dimensionality of latent codes. Therefore their bound becomes $O(m^{-\frac{1}{1.5n_x}} + m^{-\frac{1}{1.5n}})$, which is significantly worse than our bound $O(m^{-1/(n+2)} + m^{-1/(n_x+2)})$. As a result, our work provides tighter generalization bounds for both GANs and Autoencoder models. More importantly, our results for DNNs with Dropout or spectral norm translate directly to Autoencoders, leading to significant tighter bounds.

### C.3  How large is 1-upper Wasserstein dimension?

This part provides an example of why 1-upper Wasserstein dimension is not small. Before that we need to take the following definitions from Husain et al. (2019).

**Definition 3** (Covering number). *For a set $S \subset \mathbb{R}^n$, we denote $N_\eta(S)$ be the $\eta$-covering number of $S$, which is the smallest non-negative integer $m$ such that there exists closed balls $B_1, B_2, ..., B_m$ of radius $\eta$ with $S \subseteq \bigcup_{i=1}^m B_i$.*

*For any distribution $P$, the $(\eta, \tau)$-dimension is $d_\eta(P,\tau) := \frac{\log N_\eta(P,\tau)}{-\log\eta}$, where $N_\eta(P,\tau) := \inf\{N_\eta(S) : P(S) \geq 1 - \tau\}$.*

**Definition 4** (1-upper Wasserstein dimension). *The 1-upper Wasserstein dimension of distribution $P$ is*

$$d^*(P) := \inf\{s \in (2,\infty) : \limsup_{\eta\to 0} d_\eta(P, \eta^{\frac{s}{s-2}}) \leq s\}$$

Consider the simple case of the unit Gaussian distribution $P \equiv \mathcal{N}(x; 0, I)$ defined in the $n$-dimensional space $\mathbb{R}^n$. We will show that the 1-upper Wasserstein dimension of $P$ is $d^*(P) \geq 1.5n$.

First of all, we need to see the region $S$ such that $P(S) \geq 1 - \eta^{\frac{s}{s-2}}$. Since $P$ is a Gaussian, the Birnbaum-Raymond-Zuckerman inequality tells that $\Pr(||x||_2^2 \geq n\eta^{-\frac{s}{s-2}}) \leq \eta^{\frac{s}{s-2}}$. It implies that $\Pr(||x||_2^2 \leq n\eta^{-\frac{s}{s-2}}) \geq 1 - \eta^{\frac{s}{s-2}}$. In other words, $S$ is the following ball:

$$S = \{x \in \mathbb{R}^n : ||x||_2^2 \leq n\eta^{-\frac{s}{s-2}}\}$$

As a consequence, we can lower bound the covering number of $S$ as

$$N_\eta(S) \geq \left(\frac{\sqrt{n\eta^{-\frac{s}{s-2}}}}{\eta}\right)^n = \left(n\eta^{\frac{-3s+4}{s-2}}\right)^{\frac{n}{2}}$$

By definition we have

$$N_\eta(P, \eta^{\frac{s}{s-2}}) = \inf\{N_\eta(S) : P(S) \geq 1 - \eta^{\frac{s}{s-2}}\} \geq \left(n\eta^{\frac{-3s+4}{s-2}}\right)^{\frac{n}{2}}$$

Next we observe that

$$d_\eta(P, \eta^{\frac{s}{s-2}}) = \frac{1}{-\log\eta}\log N_\eta(P, \eta^{\frac{s}{s-2}}) \tag{22}$$

$$\geq \frac{1}{-\log\eta}\left[\frac{n}{2}\log n + \frac{n}{2}\left(\frac{-3s+4}{s-2}\right)\log\eta\right] \tag{23}$$

$$\geq \frac{n}{2}\left(\frac{3s-4}{s-2}\right) - \frac{n}{2\log\eta}\log n \tag{24}$$

Therefore

$$\limsup_{\eta\to 0} d_\eta(P, \eta^{\frac{s}{s-2}}) \geq \frac{n}{2}\left(\frac{3s-4}{s-2}\right) \tag{25}$$

The definition of $d^*(P)$ requires $\limsup_{\eta\to 0} d_\eta(P, \eta^{\frac{s}{s-2}}) \leq s$ and $s \in (2, \infty)$. Those requirements imply $\frac{n}{2}\left(\frac{3s-4}{s-2}\right) \leq s$, and thus $s \geq \frac{1}{4}\left(4 + 3n + \sqrt{(4+3n)^2 - 32n}\right) > \frac{3n}{2}$. As a result, $d^*(P) > 1.5n$.

## D  WHY DOES DATA AUGMENTATION IMPOSE A LIPSCHITZ CONSTRAINT?

In this section, we study a perturbed version of GANs to see the implicit role of data augmentation (DA). Consider the following formulation:

$$\min_G \max_D \mathbb{E}_{x\sim P_d}\mathbb{E}_\epsilon[\log D(x+\epsilon)] + \mathbb{E}_{z\sim P_z}\mathbb{E}_\epsilon[\log(1 - D(G(z)+\epsilon))] \tag{26}$$

where $\epsilon = \sigma u$ and $u$ follows a distribution with mean 0 and covariance matrix $I$, $\sigma$ is a non-negative constant. Note that when $u$ is the Gaussian noise, the formulation (26) turns out to be the noisy version of GAN (Arjovsky & Bottou, 2017).

*Noise penalizes the Jacobian norms:* Adding noises to the discriminator inputs corresponds to making a convolution to real and fake distributions (Roth et al., 2017; Arjovsky & Bottou, 2017). Let $p_{d*\epsilon}(x) = \mathbb{E}_\epsilon[p_d(x+\epsilon)], p_{g*\epsilon}(x) = \mathbb{E}_\epsilon[p_g(x+\epsilon)]$ be the density functions of the convoluted distributions $P_{d*\epsilon}, P_{g*\epsilon}$, respectively. We rewrite $V_\epsilon(D, G) = \mathbb{E}_{x\sim P_d}[\mathbb{E}_\epsilon\log D(x+\epsilon)] + \mathbb{E}_{z\sim P_z}[\mathbb{E}_\epsilon\log(1 - D(G(z)+\epsilon))] = \mathbb{E}_{x\sim P_{d*\epsilon}}\log D(x) + \mathbb{E}_{x\sim P_{g*\epsilon}}\log(1 - D(x))$. Given a fixed $G$, the optimal discriminator is $D^*(x) = \frac{p_{d*\epsilon}(x)}{p_{d*\epsilon}(x) + p_{g*\epsilon}(x)}$ according to Arjovsky & Bottou (2017). Training $G$ is to minimize $V_\epsilon(D^*, G)$. By using the same argument as Goodfellow et al. (2014), one can see that training $G$ is equivalent to minimizing the Jensen-Shannon divergence $d_{JS}(P_{d*\epsilon}, P_{g*\epsilon})$. Appendix D.2 shows

$$\sqrt{d_{JS}(P_{d*\epsilon}, P_g)} \leq \sqrt{d_{JS}(P_{d*\epsilon}, P_{g*\epsilon})} + \sqrt{o(\sigma)} \tag{27}$$

$$\sqrt{d_{JS}(P_d, P_{g*\epsilon})} \leq \sqrt{d_{JS}(P_{d*\epsilon}, P_{g*\epsilon})} + \sqrt{o(\sigma)} \tag{28}$$

where $o(\sigma)$ satisfies $\lim_{\sigma\to 0}\frac{o(\sigma)}{\sigma} = 0$. They suggest that for a fixed $\sigma$, minimizing $d_{JS}(P_{d*\epsilon}, P_{g*\epsilon})$ implies minimizing both $d_{JS}(P_d, P_{g*\epsilon})$ and $d_{JS}(P_{d*\epsilon}, P_g)$. The same behavior can be shown for many other GANs.

**Lemma 4.** *Let $J_x(f)$ be the Jacobian of $f(x)$ w.r.t its input $x$. Assume the density functions $p_d$ and $p_g$ are differentiable everywhere in $\mathcal{Z}_x$. For any $x \in \mathcal{Z}_x$,*
*1. $[p_{d*\epsilon}(x) - p_g(x)]^2 + O(\sigma^2) = [p_d(x) - p_g(x) + o(\sigma)]^2 + \sigma^2\mathbb{E}_u\left[u^T J_x^T(p_d)J_x(p_d)u\right]$.*
*2. $[p_d(x) - p_{g*\epsilon}(x)]^2 + O(\sigma^2) = [p_d(x) - p_g(x) - o(\sigma)]^2 + \sigma^2\mathbb{E}_u\left[u^T J_x^T(p_g)J_x(p_g)u\right]$.*
*3. $[p_{d*\epsilon}(x) - p_{g*\epsilon}(x)]^2 + O(\sigma^2) = [p_d(x) - p_g(x) + o(\sigma)]^2 + \sigma^2\mathbb{E}_u\left[u^T J_x^T(p_d - p_g)J_x(p_d - p_g)u\right]$.*

**Lemma 5.** *If $u \sim \mathcal{N}(0, I)$ then $\mathbb{E}_u\left[u^T A^T A u\right] = trace(A^T A) = ||A||_F^2$ for any given matrix $A$.*

The proof of Lemma 4 appears in Appendix D.3, while Lemma 5 comes from (Avron & Toledo, 2011; Hutchinson, 1989). Lemmas 4 and 5 are really helpful to interpret some nontrivial implications.

When training $G$, we are trying to minimize the expected norms of the Jacobians of the densities induced by $D$ and $G$. Indeed, training $G$ will minimize $d_{JS}(P_{d*\epsilon}, P_{g*\epsilon})$, and thus also minimize $d_{JS}(P_d, P_{g*\epsilon})$ and $d_{JS}(P_{d*\epsilon}, P_g)$ according to (27) and (28). Because $\sqrt{d_{JS}}$ is a proper distance, minimizing $d_{JS}(P_d, P_{g*\epsilon})$ leads to minimizing $\mathbb{E}_{x \sim p_d}[(p_d(x) - p_{g*\epsilon}(x))^2]$. Combining this observation with Lemma 4, we find that training $G$ requires both $\mathbb{E}_{x \sim p_d}[(p_d(x) - p_g(x) + o(\sigma))^2]$ and $\mathbb{E}_{x \sim p_d}\mathbb{E}_u[u^T J_x^T(p_g)J_x(p_g)u]$ to be small. As a result, $\mathbb{E}_{x \sim p_g}[|||J_x(p_g)||_F^2]$ should be small due to Lemma 5. A larger $\sigma$ encourages a smaller Jacobian norm, meaning the flatter learnt distribution. A small $\sigma$ enables us to learn complex distributions. The optimal $D^*(x) = \frac{p_{d*\epsilon}(x)}{p_{d*\epsilon}(x) + p_{g*\epsilon}(x)}$ suggests that a penalty on $J_x(p_g)$ will lead to a penalty on the Jacobian of $D$.

It is also useful to observe that adding noises to real data ($x$) only will require $d_{JS}(P_{d*\epsilon}, P_g)$ to be small, whereas adding noises to fake data ($G(z)$) only will require $d_{JS}(P_d, P_{g*\epsilon})$ to be small. Lemma 4 suggests that adding noises to real data only does not make any penalty on $p_g$. Further, if noises are used for both real and fake data, we are making penalties on both $J_x(p_g)$ and $J_x(p_d - p_g)$. Note that a small $J_x(p_d - p_g)$ implies $J_x(p_d) \cong J_x(p_g)$. As a consequence, training GAN by the loss (26) will require both the zero-order ($p_g$) and first-order ($J_x(p_g)$) informations of the fake distribution to match those of the real distribution. This is surprising. The (implicit) appearance of the first-order information of $p_d$ can help the GAN training to converge faster, due to the ability to use more information from $p_d$. On the other hand, the use of noise in (26) penalizes the first-order information of the loss, and hence can improve the generalization of $D$ and $G$, following Theorem 8.

**Connection to data augmentation:** Note that each input for $D$ in (26) is perturbed by an $\epsilon$. When $\epsilon$ has a small norm, each $x' = x + \epsilon$ is a local neighbor of $x$. Noise is a common way to make perturbation and can lead to stability for GANs (Arjovsky & Bottou, 2017). Another way to make perturbation is data augmentation, including translation, cutout, rotation. The main idea is to make another version $x'$ from an original image $x$ such that $x'$ should preserve some semantics of $x$. By this way, $x'$ belongs to the neighborhood of $x$ in some senses, and can be represented as $x' = x + \epsilon$ for some $\epsilon$. Those observations suggest that when training $D$ and $G$ from a set of original and augmented images (Zhao et al., 2020a;b), we are working with an empirical version of (26). Note that our proofs for inequalities (27, 28) and Lemma 4 apply to a larger contexts than Gaussian noise, meaning that they can apply to different kinds of data augmentation.

Some recent works (Karras et al., 2020a; Tran et al., 2021) show that data augmentation (DA) for real data only will be problematic, meanwhile using DA for both real and fake data can significantly improve GANs (Zhao et al., 2020a;b; Karras et al., 2020a). Lemma 4 agrees with those observations: DA for fake data only poses a penalty on Jacobian of $p_g$ only, while DA for real data only does no penalty on $p_g$. Differrent from prior works, Lemma 4 shows that DA for both real and fake data poses a penalty on $J_x(p_g)$ and requires $J_x(p_g) \cong J_x(p_d)$. In other words, DA requires the zero- and first-order informations of $p_g$ to match those of $p_d$, while also penalizes the first-order information of the loss for better generalization of $D$ and $G$. This is surprising.

Appendix E presents our simulation study. The results show that both DA and Gaussian noise can penalize the Jacobians of $D, G$ and the loss, hence confirming the above theoretical analysis.

### D.1  LOCAL LINEARITY

Consider a function $f : \mathbb{R}^n \to \mathbb{R}$ which is differentiable everywhere in its domain. $f$ is also called locally linear everywhere. Let $\epsilon = \sigma u$, where $u$ follows a distribution with mean 0 and covariance matrix $I$, $\sigma \geq 0$, $J_x(f)$ be the Jacobian of $f$ w.r.t its input $x$. Considering $f(x+\epsilon) = f(x+\sigma u)$ as a function of $\sigma$, Taylor's theorem allows us to write $f(x + \sigma u) = f(x) + \sigma J_x(f)u + o(\sigma)$. Therefore,

$$
\begin{aligned}
\mathbb{E}_\epsilon[f(x+\epsilon)] &= \mathbb{E}_\epsilon[f(x) + \sigma J_x(f)u + o(\sigma)] & (29) \\
&= f(x) + o(\sigma) + \sigma \mathbb{E}_u[J_x(f)u] & (30) \\
&= f(x) + o(\sigma), & (31)
\end{aligned}
$$

where we have used $\mathbb{E}_u[J_x(f)u] = 0$ due to $\mathbb{E}_u[u] = 0$ and the independence of the elements of $u$. As $\sigma \to 0$, we have $\mathbb{E}_\epsilon[f(x+\epsilon)] \to f(x)$.

## D.2 PROOFS FOR INEQUALITIES (5, 6)

Consider the Jensen-Shannon divergence $d_{JS}(P_{d*\epsilon}, P_{g*\epsilon})$. Since $\sqrt{d_{JS}}$ is a proper distance, we have the following triangle inqualities:

$$\sqrt{d_{JS}(P_{d*\epsilon}, P_g)} \le \sqrt{d_{JS}(P_{d*\epsilon}, P_{g*\epsilon})} + \sqrt{d_{JS}(P_{g*\epsilon}, P_g)} \tag{32}$$

$$\sqrt{d_{JS}(P_d, P_{g*\epsilon})} \le \sqrt{d_{JS}(P_{d*\epsilon}, P_{g*\epsilon})} + \sqrt{d_{JS}(P_{d*\epsilon}, P_d)} \tag{33}$$

Next we will show that $d_{JS}(P_{g*\epsilon}, P_g) = o(\sigma)$. The following expression comes from a basic property of Jensen-Shannon divergence:

$$d_{JS}(P_{g*\epsilon}, P_g) = H\left(\frac{P_{g*\epsilon} + P_g}{2}\right) - \frac{1}{2}H(P_{g*\epsilon}) - \frac{1}{2}H(P_g), \tag{34}$$

where $H(P)$ denotes the entropy of distribution $P$.

Denote $o(\cdot), o_1(\cdot), o_2(\cdot), o_3(\cdot)$ be some functions of $\sigma$ satisfying $\lim_{\sigma \to 0} \frac{o(\sigma)}{\sigma} = 0$. Appendix D.1 suggests that $p_{g*\epsilon}(x) = \mathbb{E}_\epsilon[p_g(x+\epsilon)] = p_g(x) + o_1(\sigma)$ and $\log(p_g(x) + o_1(\sigma)) = \log(p_g(x)) + o_2(\sigma)$ by using Taylor's theorem for $\sigma$. Therefore:

$$-\frac{1}{2}H(P_g) = \frac{1}{2}\int p_g(x)\log p_g(x)dx. \tag{35}$$

$$\begin{aligned} -\frac{1}{2}H(P_{g*\epsilon}) &= \frac{1}{2}\int p_{g*\epsilon}(x)\log p_{g*\epsilon}(x)dx = \frac{1}{2}\int p_{g*\epsilon}(x)\log[p_g(x) + o_1(\sigma)]dx \\ &= \frac{1}{2}\int p_{g*\epsilon}(x)[\log p_g(x) + o_2(\sigma)]dx \\ &= \frac{1}{2}\int p_{g*\epsilon}(x)\log p_g(x)dx + \frac{1}{2}o_2(\sigma). \end{aligned} \tag{36}$$

$$\begin{aligned} H\left(\frac{P_{g*\epsilon} + P_g}{2}\right) &= -\int \frac{p_{g*\epsilon}(x) + p_g(x)}{2}\log\left(\frac{p_{g*\epsilon}(x) + p_g(x)}{2}\right)dx \\ &= -\int \frac{p_{g*\epsilon}(x) + p_g(x)}{2}\log\left(\frac{2p_g(x) + o_1(\sigma)}{2}\right)dx \\ &= -\int \frac{p_{g*\epsilon}(x) + p_g(x)}{2}\log\left(p_g(x) + \frac{1}{2}o_1(\sigma)\right)dx \\ &= -\int \frac{p_{g*\epsilon}(x) + p_g(x)}{2}\left[\log p_g(x) + \frac{1}{2}o_3(\sigma)\right]dx \\ &= -\frac{1}{2}\int [p_{g*\epsilon}(x) + p_g(x)]\log p_g(x)dx - \frac{o_3(\sigma)}{4}\int [p_{g*\epsilon}(x) + p_g(x)]dx \\ &= -\frac{1}{2}\int [p_{g*\epsilon}(x) + p_g(x)]\log p_g(x)dx - \frac{1}{2}o_3(\sigma) \\ &= -\frac{1}{2}\int p_{g*\epsilon}(x)\log p_g(x)dx - \frac{1}{2}\int p_g(x)\log p_g(x)dx - \frac{1}{2}o_3(\sigma) \end{aligned} \tag{37}$$

From equations (34, 35, 36, 37) we can conclude $d_{JS}(P_{g*\epsilon}, P_g) = o(\sigma)$. Similar arguments can be done to prove $d_{JS}(P_{d*\epsilon}, P_d) = o(\sigma)$. Combining those with (32) and (33), we arrive at

$$\sqrt{d_{JS}(P_{d*\epsilon}, P_g)} \leq \sqrt{d_{JS}(P_{d*\epsilon}, P_{g*\epsilon})} + \sqrt{o(\sigma)} \tag{38}$$

$$\sqrt{d_{JS}(P_d, P_{g*\epsilon})} \leq \sqrt{d_{JS}(P_{d*\epsilon}, P_{g*\epsilon})} + \sqrt{o(\sigma)}. \tag{39}$$

### D.3 Proof of Lemma 13

For any $x \in \mathcal{Z}_x$, it is worth remembering that $p_{d*\epsilon}(x) = \mathbb{E}_\epsilon[p_d(x+\epsilon)]$. Consider
$$Y = p_d(x+\epsilon) - p_g(x) = p_d(x+\sigma u) - p_g(x),$$
which is a function of $u$. Since $u$ follows a distribution with mean 0 and covariance which is the identity matrix, $Y$ is a random variable. Due to $p_d(x+\epsilon) = p_d(x) + \sigma J_x(p_d)u + o(\sigma)$ from Appendix D.1, we can express the variance of $Y$ as

$$
\begin{aligned}
\mathrm{Var}(Y) &= \mathbb{E}_u(Y^2) - (\mathbb{E}_u(Y))^2 \tag{40} \\
&= \mathbb{E}_u\left[(p_d(x+\epsilon) - p_g(x))^2\right] - [\mathbb{E}_u(p_d(x+\epsilon) - p_g(x))]^2 \tag{41} \\
&= \mathbb{E}_u\left[[p_d(x) - p_g(x) + o(\sigma) + \sigma J_x(p_d)u]^2\right] - [p_{d*\epsilon}(x) - p_g(x)]^2 \tag{42} \\
&= \mathbb{E}_u\left[(p_d(x) - p_g(x) + o(\sigma))^2\right] + 2\sigma[p_d(x) - p_g(x) + o(\sigma)]\mathbb{E}_u[J_x(p_d)u] \\
&\quad + \sigma^2\mathbb{E}_u\left[u^T J_x^T(p_d)J_x(p_d)u\right] - [p_{d*\epsilon}(x) - p_g(x)]^2 \tag{43} \\
&= [p_d(x) - p_g(x) + o(\sigma)]^2 + \sigma^2\mathbb{E}_u[u^T J_x^T(p_d)J_x(p_d)u] - [p_{d*\epsilon}(x) - p_g(x)]^2 \tag{44}
\end{aligned}
$$

Since $p_g(x)$ does not depend on $\epsilon = \sigma u$, we have $\mathrm{Var}(Y) = \mathrm{Var}(p_d(x+\epsilon)) = \mathrm{Var}(p_d(x+\sigma u))$ which is bounded above by $C\mathrm{Var}(\sigma u) = C\sigma^2$, for some $C \geq 0$. Combining this with (44) will result in

$$[p_{d*\epsilon}(x) - p_g(x)]^2 + C\sigma^2 \geq [p_d(x) - p_g(x) + o(\sigma)]^2 + \sigma^2\mathbb{E}_u[u^T J_x^T(p_d)J_x(p_d)u] \tag{45}$$

completing the first statement. The second and third statements can be proven similarly.

## E Evaluation of data augmentation for GANs

There is a tradeoff in data augmentation. Making augmentation from a larger region around a given image implies a larger $\sigma$. Lemmas 13 and 14 suggest that the Jacobian norms should be smaller, meaning the flatter learnt distributions. Hence, too large region for augmentation may result in underfitting. On the other hand, augmentation in a too small region (a small $\sigma$) allows the Jacobian norms to be large, meaning the learnt distributions can be complex. As $\sigma \to 0$, no regularization is used at all.

This section will provide some empirical evidences about those analyses. We first evaluate the role of $\sigma$ when doing augmentation by simple techniques such as translation. We then evaluate the case of augmentation by adding noises. Two models are used in our evaluations: Saturating GAN (Goodfellow et al., 2014) and LSGAN (Mao et al., 2017).

### E.1 Experimental setups

The architectures of $G$ and $D$ are specified in Figure 1, which follow `http://github.com/eriklindernoren/PyTorch-GAN/blob/master/implementations/gan/gan.py`. We use this architecture with Spectral normalization (Miyato et al., 2018) for $D$ in all experiments of GAN and LSGAN. Note that, for LSGAN, we remove the last Sigmoid layer in $D$.

We use MNIST dataset which has 60000 images for training and 5000 images for testing. During the testing phase, 5000 new noises are sampled randomly at every epoch/minibatch to compute some metrics. For the derivative of $D$ with respect to its input, the input includes 2500 fake images and 2500 real images. Before fetching into $D$, both real and fake images are converted to tensor size $(1, 28, 28)$, rescaled to $(0, 1)$ and normalized with $mean = 0.5$ and $std = 0.5$. The noise input of $G$ has 100 dimensions and is sampled from normal distribution $\mathcal{N}(0, I)$. We use Adam optimizer with $\beta_1 = 0.5, \beta_2 = 0.999, lr = 0.0002, batchsize = 64$.

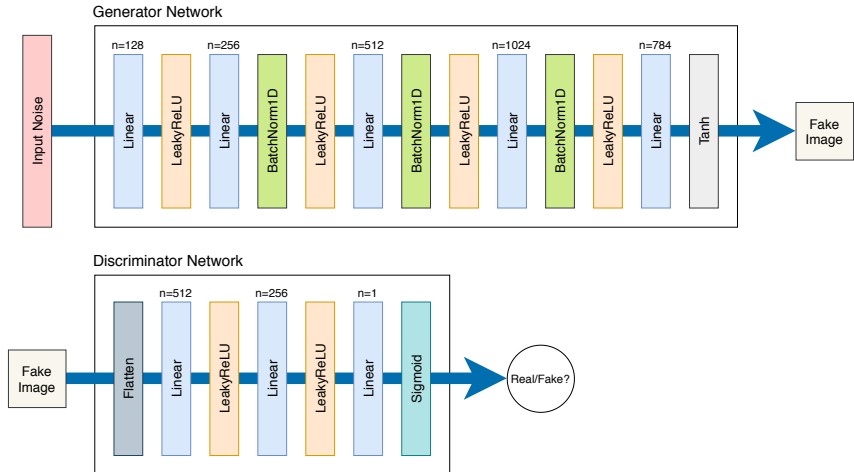

Figure 1: The architectures of $G$ and $D$ with the negative slope of `LeakyRuLU` is $0.2$

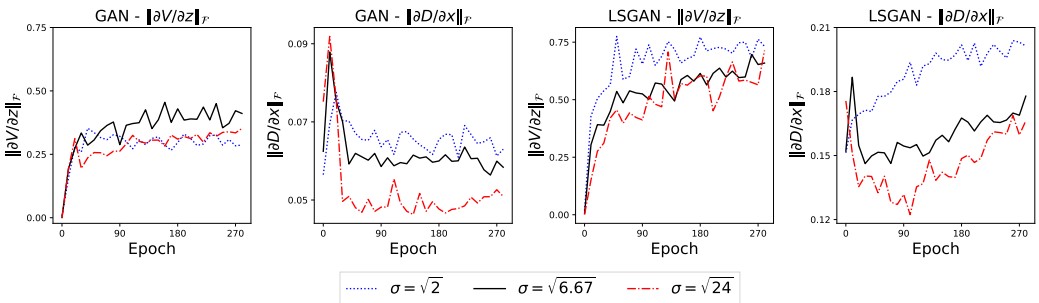

Figure 2: Some behaviors of GAN (first two subfigures) and LSGAN (last two subfigures) with different $\sigma$ for augmentation. Both $\frac{\partial D}{\partial x}$ and $\frac{\partial V}{\partial z}$ are measured along the training process. $\|\cdot\|_F$ denotes the Frobenious norm.

### E.2 THE ROLE OF $\sigma$ FOR DATA AUGMENTATION

In this experiment, the input of $D$ which includes real and fake images are augmented using translation. The shifts in horizontal and vertical axis are sampled from discrete uniform distribution within interval $[-s, s]$, where $s = 2$ corresponds to $\sigma = \sqrt{2}$, $s = 4$ corresponds to $\sigma = \sqrt{6.67}$, and $s = 8$ corresponds to $\sigma = \sqrt{24}$.

*Jacobian norms of $D$ and loss $V$:* Figure 2 shows the results. It can be seen from the figure that the higher $\sigma$ provides smaller Frobenius norms of Jacobian of both $D$ and $V$. Such behaviors appear in both GAN and LSGAN, which is consistent with our theory.

*Jacobian norms of $G$:* To see the effect of data augmentation on $G$, we need to fix $D$ when training $G$. Therefore we did the following steps: (i) train both $G$ and $D$ for 100 epochs, (ii) then keeping $D$ fixed, we further train $G$ to measure its Jacobian norm along the training progress. We chose $\sigma \in \{\sqrt{6.67}, \sqrt{14}, \sqrt{24}\}$ and augmented $64, 96, 128$ times for each image respectively.

Figure 3 shows the results. We observe that a higher $\sigma$ provides smaller Jacobian norm of $G$. Interestingly, as the norm decreases as training $G$ more, suggesting that $G$ gets simpler.

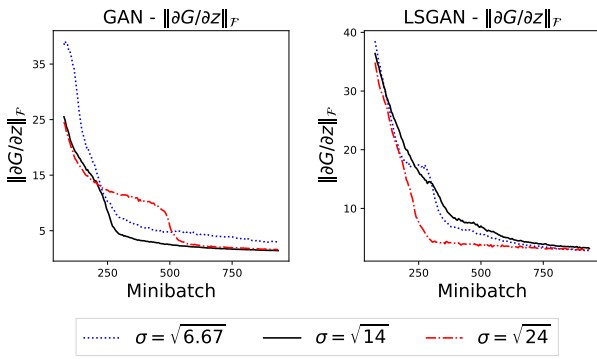

Figure 3: Some behaviors of GAN and LSGAN with different $\sigma$ for augmentation. $\frac{\partial G}{\partial z}$ is measured along the training process.

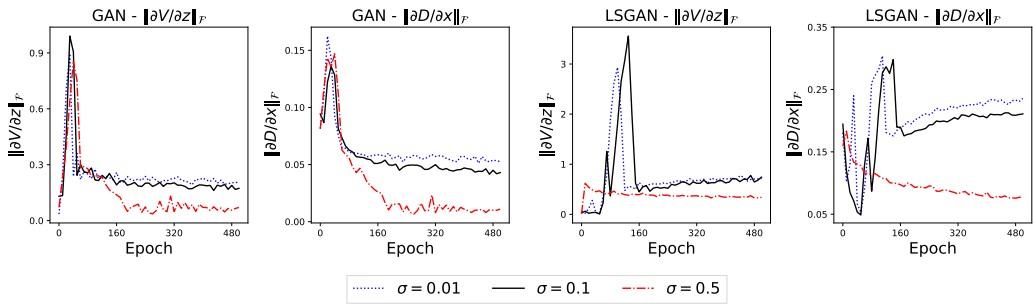

Figure 4: Some behaviors of GAN (first two subfigures) and LSGAN (last two subfigures) when augmenting images by adding noises. Both $\frac{\partial D}{\partial x}$ and $\frac{\partial V}{\partial z}$ are measured along the training process.

### E.3 AUGMENTATION BY ADDING NOISES

In this experiment, the input of $D$ which includes real and fake images are augmented by adding Gaussian noise $\mathcal{N}(0, \sigma)$. We choose $\sigma \in \{0.01, 0.1, 0.5\}$ and augment $\{16, 64, 128\}$ times for each image respectively.

*Jacobian norms of D and loss V:* Figure 4 shows the results after 500 epochs. It can be seen from the figure that the higher $\sigma$ provides smaller Jacobian norms. This is consistent with our theoretical analysis. In comparison with using translation, adding Gaussian noise makes the Jacobian norms in both GAN and LSGAN more stable.

*Jacobian norms of G:* We did the same procedure as for the case of image translation to see how large the norm of $\partial G/\partial z$ is. We choose $\sigma \in \{0.5, 2, 4\}$. Figure 5 show the results. The same behaviour can be observed. Larger $\sigma$ often leads to smaller norms. It is worth noting that the Jacobian norm will be zero as $\sigma$ is too large. In this case both $D$ and $G$ may be over-penalized. Those empirical results support well our theory.

## F EVALUATION OF SPECTRAL NORMALIZATION

This section presents an evaluation on the effect of Lipschitz constraint by using spectral normalization (SN) (Miyato et al., 2018). We use Saturating GAN and LSGAN with four scenerios: no penalty; SN for $G$ only; SN for $D$ only; SN for both $D$ and $G$. The setting for our experiments appears in subsection E.1.

The results appear in Figure 6. When no penalty is used, we observe that the gradients of the loss tend to increase in magnitude while both $D$ and $G$ are hard to reach optimality. When SN is used

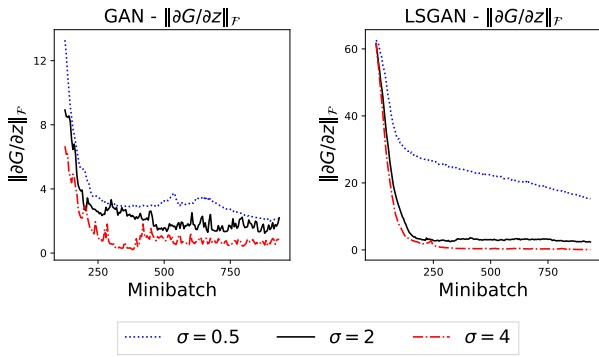

Figure 5: Some behaviors of GAN and LSGAN when augmenting images by adding noises. $\frac{\partial G}{\partial z}$ is measured along the training process.

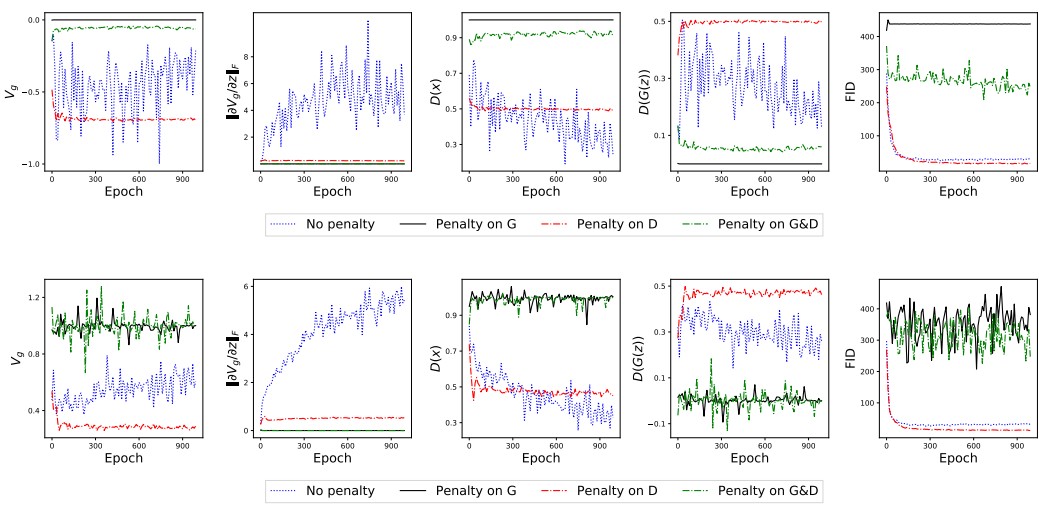

Figure 6: Some behaviors of Saturating GAN (top row) and LSGAN (bottom row) in different situations. $V_g$ is the loss for training the generator, and FID measures the quality of generated images, the lower the better.

for $G$, it seems that $G$ has been over-penalized since the gradient norms are almost zero, meaning that $G$ may be underfitting. This behavior appears in both GAN and LSGAN, and was also observed before (Brock et al., 2019). The most sucessful case is the use of SN for $D$ only. We observe that both players seem to reach the Nash equilibrium. The gradient norms of the loss are relatively stable and small in the course of training, while the quality of fake image (measured by FID) can be better than the other cases. Furthermore both the loss $V_g$ and $\|\partial V_g/\partial z\|_F$ of the generator are stable and belong to small domains, suggesting that the use of SN for $D$ can help us to penalize the zero- and first-order informations of the loss.

Our experiments suggest three messages which agree well with our theory in Theorem 8. Firstly, when no penalty is used, the Lipschitz constant of a hypothesis may be large in order to well fit the training data. In this case the generalization may not be good. Secondly, we can get stuck at underfitting if a penalty on Lipschitzness is overused. The reason is that a heavy penalty can result in a small Lipschitz constant (thus simpler hypothesis), meanwhile a too simple hypothesis may cause a large optimization error. Hence, the generalization is not good in this case. Thirdly, when an appropriate penalty is used, we can obtain both a small Lipschitz constant and small optimization error which lead to better generalization.

## G   FURTHER DISCUSSION

We have discussed both generalization and consistency in Section 3. We next provide some interpretations from our theoretical results which may be helpful in practice.

- We may want to find an unknown (measurable) function $\eta(x)$ based on a training set $\boldsymbol{S}$ of size $m$.[3] A popular way is to select a family $\mathcal{H}$ (e.g., an NN architecture) and then do training on $\boldsymbol{S}$ to obtain a specific $h_o \in \mathcal{H}$. The quality of $h_o$ can be seen from different levels (Bousquet et al., 2004):

  $E_1$. *Optimization error:* $err_o(h_o, h_m^*) = F(\widehat{P}_x, h_o) - F(\widehat{P}_x, h_m^*)$ for comparing with $h_m^* = \arg\min_{h \in \mathcal{H}} F(\widehat{P}_x, h)$ which is the best in $\mathcal{H}$ for the training data;

  $E_2$. *Generalization gap:* $err_g(h_o) = |F(P_x, h_o) - F(\widehat{P}_x, h_o)|$ to see the difference between the empirical and expected losses of $h_o$;

  $E_3$. *Consistency rate:* $err_c(h_o, h^*) = F(P_x, h_o) - F(P_x, h^*)$ for comparing with the best function $h^* = \arg\min_{h \in \mathcal{H}} F(P_x, h)$ in $\mathcal{H}$;

  $E_4$. *Bayes gap:* $err_B(h_o, \eta) = F(P_x, h_o) - F(P_x, \eta)$ for comparing with the truth.

- A small optimization error may not always lead to good generalization.

- A small generalization gap is insufficient to explain a high success in practice. Indeed, $err_g(h_o)$ can be small although both empirical and expected losses are high. The use of this quantity poses a long debate (Nagarajan & Kolter, 2019; Negrea et al., 2020).

- From Theorem 1, one may try to penalize the Lipschitz constant of the loss as small as possible to ensure a small generalization gap. However, as explained before, such a naive application may not lead to good performance. The reason is that family $\mathcal{H}$ may be much smaller and the members of $\mathcal{H}$ will have lower capacity as $L$ decreases. Note that a large decrease of capacity easily leads to underfitting, and hence $F(P_x, h_o)$ will be high. Our experiments in Appendix F provide a further evidence when spectral normalization is overused.

- Those observations suggest that making only optimization or generalization gap small is not enough. Both should be small, and so is consistency rate due to Lemma 1.

- *When does a small consistency rate still lead to bad generalization?* In those bad cases, the Bayes gap $err_B(h_o, \eta)$ will be large. Note that $err_B(h_o, \eta) = err_c(h_o, h^*) + err_a(\mathcal{H})$, where $err_a(\mathcal{H}) = F(P_x, h^*) - F(P_x, \eta)$ is often known as the approximation error and measures how well can functions in $\mathcal{H}$ approach the target (Bousquet et al., 2004). Therefore $err_a(\mathcal{H})$ represents the capacity of family $\mathcal{H}$. A stronger family with higher-capacity members will lead to smaller $F(P_x, h^*)$ and hence a smaller $err_a(\mathcal{H})$. Those observations imply that, provided loss $f$ is not a constant function, a bad generalization with a small consistency rate happens only when $\mathcal{H}$ has low capacity.

- *When working with a high-capacity family $\mathcal{H}$, a small consistency rate is sufficient to ensure good generalization. Lemma 1 suggests that it is sufficient to ensure good generalization by making both optimization error and generalization gap to be small.*

- For overparameterized NNs, we often observe small (even zero) optimization error. Our results in Theorem 3 shows that Dropout and spectral normalization can produce small generalization gap. By combining those observations, we can conclude that Dropout DNNs and SN-DNNs can generalize well.

---

[3]For simplicity, we limit the discussion to measurable functions.

