# OpenReview forum: "Generalization of GANs and overparameterized models under Lipschitz continuity"
_ICLR.cc/2022/Conference — ICLR 2022 Submitted_

### Official Review · Reviewer_wKt9 · 2021-11-02

**Correctness:** 3
**Technical Novelty And Significance:** 2
**Empirical Novelty And Significance:** Not applicable
**Recommendation:** 3
**Confidence:** 4

**Main Review:**

Generative adversarial networks (GANs) usually achieve satisfactory generalization performance in practice, although their optimization problems are solved over deep neural network models with large capacities. This paper attempts to understand how the generalization error will depend on the Lipschitz constant of the neural networks in GANs. To reach this goal, the paper shows Theorem 1 bounding the generalization error with the Lipschitz constant of the neural network function (L) and the diameter of the learned variables B. Then, the paper applies this result to the GAN problem and connects the assumption on the Lipschitz constant to spectral normalization and dropout regularization techniques.

The paper studies a highly interesting theoretical problem to understand the success of Lipschitz regularization methods. However, I think the shown bounds are too conservative and do not justify the generalization success in GANs. This is because the generalization result in Theorem 1 aims to bound the generalization error uniformly over the entire space of 1-Lipschitz functions. Consequently, the bounds are all exponentially growing with the dimension of data vector (n) and become vacuous for moderately high-dimensional data vectors with a dimension greater than 20 or so. Therefore, the bounds do not address the curse of dimensionality in deep learning experiments and only show the generalization error is bounded with an exponential function of dimension n which does not apply to practical deep learning settings.  On the other hand, the generalization bound in (Arora et al 2017) does not exponentially grow with data dimension through bounding the number of variables of the neural net players as a specification for the GAN problem.  Also, the paper does not provide any numerical evaluation of the shown bounds and how they change with the actual generalization error in GANs. Due to these reasons, I do not recommend this paper for publication in its current form. The paper's generalization bounds can be significantly improved by taking the optimization algorithm and the design of neural network players into account, and I suggest the authors improve their results in that way.

**Summary Of The Paper:**

The paper tries to bound the generalization error of deep learning models and generative adversarial networks with the Lipschitz coefficient of the model. The main idea supported by the paper's discussion is that bounding the Lipschitz constant of the neural networks will lead to good generalization performance. The paper observes the connection between spectral normalization and norm-bounded dropout with the Lipschitz coefficient of the neural network and extends the proven bounds to those regularization methods.

**Summary Of The Review:**

While the paper proves several generalization bounds, its main result (Theorem 1) models a neural network by only its Lipschitz constant and tries to bound the generalization error uniformly over all 1-Lipschitz functions. Therefore, the bounds grow exponentially with the data dimension and lose their power for even moderately large data vectors including all practical deep learning settings. Therefore, I think the paper's analysis does not help with understanding generalization in GANs and needs to be significantly improved by considering the design of neural net players and the optimization algorithm.

---

> ### Author Response · Authors · 2021-11-20
> **Responses to Reviewer wKt9 (2)**
>
>
> **3. Also, the paper does not provide any numerical evaluation of the shown bounds and how they change with the actual generalization error in GANs.**
>
> This is a limitation of our work. However, we provided various evaluations about the effects of spectral normalization (in Appendix F) and data augmentation (in Appendix E). Those experiments reflect clearly the role of SN and data augmentation for regularizing the players and show important evidences for our theoretical analyses. We believe those evaluations are more important than an empirical evaluation of a theoretical bound. The reason is that our bounds may hide some constants that had not been tighten.
>
>
> **4. The paper's generalization bounds can be significantly improved by taking the optimization algorithm and the design of neural network players into account, and I suggest the authors improve their results in that way.**
>
> Thank you very much for this suggestion. We did employ the design of NNs in our analyses. Indeed we used Dropout and spectral normalization to show important results that answer three long-standing questions of deep learning. Those results appears in Theorems 2, 3, 7, 9, and Corollary 2. Analyzing more specialized architectures may requires entirely new studies, and we leave it open for future research.
>
> Optimization is another way to improve our bounds. This work focuses on capacity, decomposes quality (in terms of consistency) of a function into "optimization error + generalization error", and provides efficient bounds for "generalization error". We believe that such a decomposition opens a large door for future research. A new advancement in optimization can be inherited directly to improve consistency results for DNNs. Therefore, instead of a limitation, our presentation is really beneficial for future study.
>
>
> ### A final remark:
> We urge Reviewer wKt9 to read our paper again and try to understand the main contributions which provide a significant step toward answering the open theoretical issues of deep learning. We will really appreciate if the reviewer update his/her score to better reflect our contributions.

---

> ### Author Response · Authors · 2021-11-20
> **Responses to Reviewer wKt9 (1)**
>
>
> We would like to thank the reviewer's time and effort for providing feedbacks. However, we must respectfully disagree with the reviewer's main opinions. It seems that the reviewer seriously misunderstood the role of Theorem 1 and missed all remaining parts which contain our most significant contributions for DNNs and GANs.
>
> In the following, we will point out the main misunderstandings in the review.
>
> **1. I think the shown bounds are too conservative and do not justify the generalization success in GANs. This is because the generalization result in Theorem 1 aims to bound the generalization error uniformly over the entire space of 1-Lipschitz functions. Consequently, the bounds are all exponentially growing with the dimension of data vector (n) and become vacuous for moderately high-dimensional data vectors with a dimension greater than 20 or so. Therefore, the bounds do not address the curse of dimensionality in deep learning experiments and only show the generalization error is bounded with an exponential function of dimension n which does not apply to practical deep learning settings.**
>
> Theorem 1 in Section 3 is not for a family of 1-Lipschitz functions only. Instead, it presents the main connection between Lipschitz continuity and generalization in a very general setting. Such a connection is crucial for our later analyses for deep neural networks (DNNs) and GANs. Since little information of family $H$ is assumed, it is natural to see that the bounds in Theorem 1 suffer from the curse of dimensionality. We already discussed this limitation in the paragraph just before subsection 3.1.
>
> Reviewer wKt9 should read subsections 3.1 and 3.2 to see why the curse of dimensionality can be avoided.
>  - We show in Theorem 3 that Dropout and spectral normalization (SN) can help DNNs generalize better and avoid the curse of dimensionality.
>  - We show that deeper NNs will provide better generalization error.
>  - We analyze both generalization error and consistency to assess the goodness of a learnt function. We repeatedly remind (e.g., before subsection 3.2 and below equation 2) that using generalization error only cannot explain a high performance. Therefore one needs consistency.
>  - Corollary 2 shows that DNNs with Dropout or SN can has small consistency rate which does not suffer from the curse of dimensionality.
>
> Note that Dropout is prevalent in machine learning while SN is used popularly in GANs. For the first time in the literature, our results provide answers for the two long-standing questions in deep learning:
>  - *Why deeper NNs can generalize better?*
>  - *Whether or not the generalization of deep NNs suffers from the curse of dimensionality?*
>
>
> **2. On the other hand, the generalization bound in (Arora et al 2017) does not exponentially grow with data dimension through bounding the number of variables of the neural net players as a specification for the GAN problem.**
>
> We agree that their bound may not suffer from the curse of dimensionality. Nonetheless, as pointed out in Section 2, their analyses and bounds using neural distance have various limitations and may not reflect generalization in traditional senses. For examples,
>
> - They provided upper bounds for quantity $| d_{\mathcal{D}}(P_d, P_g) - d_{\mathcal{D}}( \widehat{P}_d, \widehat{P}_g)|$ to see the goodness of a learnt distribution $P_g$. For a suitable choice of the loss $V(P_d, P_z, D, G)$ in GANs, we can write $| d_D (P_d, P_g) - d_D( \widehat{P}_d, \widehat{P}_g) | = | \max_D V(P_d, P_z, D, G) - \max_D V(\widehat{P}_d, \widehat{P}_z, D, G) |$, where   $P_g$ is the induced distribution by putting samples from $P_z$ through generator $G$. Such a quantity is non-traditional due to using the best $D$.
>
> - Given a $G$, let $D_1 = \arg \max_D V(P_d, P_z, D, G)$ and $D_2 = \arg \max_D V(\widehat{P}_d, \widehat{P}_z, D, G)$. Note that $D_1$ and $D_2$ need not to be the same. Therefore, a bound on $| d_D (P_d, P_g) - d_D ( \widehat{P}_d, \widehat{P}_g) | = | V(P_d, P_z, D_1, G) - V(\widehat{P}_d, \widehat{P}_z, D_2, G) | $ *does not reflect generalization error of neither a specific $G$ nor $(D, G)$.*
>
> - For any given $G$, $| d_{\mathcal{D}}(P_d, P_g) - d_{\mathcal{D}}( \widehat{P}_d, \widehat{P}_g)|$ will be smaller as increasing the capacity of the family that defines the neural distance.
>
> - The neural distance between two given distributions $(\mu, \nu)$  may be small even when the two are far away (Arora et al., 2017). This is because there exists a perfect discriminator $D$, whenever $\mu$ and $\nu$ do not have overlapping supports (Arjovsky \& Bottou, 2017). In those cases, a distance-based bound may be trivial.
>
> Those reasons suggest that one cannot use their bounds to tell whether or not generalization of GANs can avoid the curse of dimensionality.

---

> > ### Comment · Reviewer_wKt9 · 2021-11-29
> > **Response to Authors' Rebuttal**
> >
> > I read the authors' response and carefully checked Theorem 3 which is the paper's central result on the generalization of neural networks with spectral regularization and dropout. The authors claim that the bound in this theorem does not suffer from the curse of dimensionality.  However, based on the proof of Theorem 3, it seems to me that Theorem 3 is a trivial result that deterministically holds for every neural network with a sufficiently small Lipschitz constant.
> >
> > To be more precise about my point, let me refer to the proof of Theorem 3 on page 17. The first inequality in this proof says:
> >
> > $$\sup_{h\in H_{dr}} | F(P_x,h) - F(\widehat{P}_X,h)| \le BL + C\sqrt{(\log 4 -2\log\delta )/m}$$
> >
> > I claim that one can easily see the second term in the above inequality ($C\sqrt{(\log 4 -2\log\delta )/m}$) is actually not needed, and we always have the following trivial inequality which only uses the simple fact that for a diameter $B$ (where we have $\Vert \mathbf{x} -  \mathbf{x}' \Vert\le B$) and $L$-Lipschitz function $F$ for any feasible $x,x'$ we have $|F(x,h) -  F(x',h)|\le LB$:
> >
> > $$\sup_{h\in H_{dr}} | F(P_x,h) - F(\widehat{P}_X,h)| \le BL $$
> >
> > Then, the paper's theorem 3 seems to suppose an extremely powerful spectral or dropout regularization to make sure the Lipschitz constant satisfies $L\le L_f\frac{C_{dr}}{\sqrt{m}}$ with $m$ denoting the sample size. Of course, under such a heavy regularization one can see for every two points $\mathbf{x},\mathbf{x}'$ we have the following:
> >
> > $$\forall x,x': |F(x,h) -  F(x',h)|\le BL \le BL_f\frac{C_{dr}}{\sqrt{m}} \Rightarrow  \sup_{h\in H_{dr}} | F(P_1,h) - F(P_2,h)| \le BL \le BL_f\frac{C_{dr}}{\sqrt{m}}$$
> >
> > which says the result would trivially hold for every two distributions $P_1,P_2$ and does not have any non-trivial implications about the generalization of regularized neural networks. As I said above, the above argument suggests that the paper's claims regarding the generalization bounds for spectrally regularized neural networks do not have any non-trivial implications and I would rather keep my original score.

---

> > > ### Author Response · Authors · 2021-11-29
> > > **Response for new arguments**
> > >
> > > Thank you for providing further arguments. However, **your arguments are nonlogical**:
> > >
> > > - Consider a family $H$ with members having sufficiently small Lipschitz constants, you can show a small generalization gap for $H$. Note that one cannot conclude a similar generalization gap for Dropout DNNs, since $H$ does not necessarily contain those Dropout DNNs.
> > > Your direct translation from a trivial result for $H$ into Dropout DNNs is thus nonlogical and misleading. This is important  especially for the knowledge that no existing paper shows a small Lipschitz constant for Dropout DNNs.
> > >
> > > - Our findings about Dropout DNNs and SN-DNNs are **nontrivial**, since for the first time in the literature we show in Theorem 2 that those DNNs have small Lipschitz constants which can be exponentially smaller as the network depth increases. Those findings are especially meaningful for the cases of overparameterized DNNs which have high capacity. The assumptions we used are very practical as explained in the paragraphs after Theorems 2 and 3. Please correct us if this is wrong!
> > >
> > > A small generalization gap in Theorem 3 will lead to a small consistency rate in Corollary 2. Such a guarantee on small consistency rate is really crucial to ensure a high performance of overparameterized Dropout DNNs as pointed out in the last two paragraphs of Section 3. Those observations suggest that our small generalization bounds for Dropout DNNs and SN-DNNs have nontrivial implications.

---

### Official Review · Reviewer_wjLb · 2021-11-02

**Correctness:** 3
**Technical Novelty And Significance:** 3
**Empirical Novelty And Significance:** Not applicable
**Recommendation:** 5
**Confidence:** 4

**Main Review:**

**Writing in the abstract and introduction:**

The introduction and abstract are not written well and need several revisions before the paper is fit for publication. This is one of the biggest weaknesses of the paper. I will provide three examples here of areas of improvement. Specifically, the introduction does not do the paper justice and can easily turn off a reader before they even get to the bulk of the work (which is interesting and has value).
1) The whole abstract is very vague and doesn’t explain much. The first sentence is not saying almost anything at all and the last sentence mentions “the long mystery of why imposing a Lipschitz constraint can help GANs to generalize well in practice” offhandedly. The last sentence was actually very confusing to me at first because plenty of work discusses Lipschitz constraints on GANs, so initially I thought that the authors were not aware of these (not-so-obscure) papers. However, after reading the related works I understood that the authors meant something more like “although experimentally Lipschitz constraints have been used with great success, the relationship between imposing a Lipschitz constraint and GAN generalization is not theoretically understood”. Whatever the authors choose to say, it should be really clear what they mean, ESPECIALLY in the abstract and introduction.
2) There are also several typos and things that are worded strangely. For example, in the second paragraph it is written “... two players competing each other.” This should be “... competing against each other.”
3) The introduction does very little to explain the actual problem in the context of literature. The authors say that “little has been known about the generalization of the trained players” and “The standard learning theories still lack an efficient tool to analyze GANs”. These are too vague for a reader to understand what THIS paper is addressing specifically.

**Specificity and confusing things:**

The authors lack specificity in several areas. Without being specific, the paper is confusing and the reader must guess what is going on. I will provide three examples here of the lack of specificity. The following are all weaknesses:

1) In the abstract itself, the authors say that GANs are “complex” in their first sentence. This is true in many ways so it is unclear how the authors address this. They say that they use efficient tools, but what does efficiency even mean here? In other words, in the abstract it isn’t clear how the work done is addressing actual open problems because of a lack of specificity.
2) In the first paragraph of the paper, they introduce the latent distribution z. In section 3, they call the data distribution (as far as I can tell) z, which is confusing. If they want to use z to represent both the latent variable and the data, then they should explicitly say that they are switching notation, although they should just stick to one.
3) The theorems and lemmas are labeled together; for example, Theorem 1, Theorem 2, Theorem 3, Lemma 4, then Theorem 5 without having Lemmas 1-3 and 5 and not having Theorem 4. However, the corollaries have independent numbering, which is confusing when trying to find things. Stick to one convention.

**Theorem 1:**

Many of the assumptions are valid, which is a big strength. For example, it is reasonable that Z is bounded by B as image datasets typically lie in the hypercube.
However a few of the assumptions are not representative for GANs in practice, which is a weakness. It doesn’t seem reasonable that the loss is bounded, as most losses are not. The typical GAN loss stated in this paper is not bounded and hence Theorem 1 does not apply. The Wasserstein loss and f-divergences in general need not be bounded. So, the authors need to justify this hypothesis.
Regardless, this result is very interesting and a great contribution (strength). The result that you can generalize better with simpler functions seems to go against the intuition that a highly complex hypothesis class is beneficial. Although this raises the important question… Since the generalization error is based on the loss f (and F), are we just generalizing better because we are making BOTH $F(P_z, h)$ and $F(\hat P_z, h)$ less able to distinguish differences? For example, clearly if we choose f = 0 then it is 0-Lipschitz and we get “perfect generalization” but obviously this isn’t good. This is a weakness until it is explained.

**Theorem 2 and 3:**

Theorem 3 is great and reasonable (strength). The authors do a great job showing how dropout and spectral normalization can be used to get better generalization bounds. It seems to me that Theorem 2 might be better renamed as a proposition because it isn’t central and not very hard to prove. But that’s just my opinion.

**Section 3.2:**

In Theorem 5 and Corollary 1, it is not clear that these models are consistent, which is a weakness. This needs to be cleared up because that is the claim of this section. Obviously as $m \rightarrow \infty$ we have that $2(LB + 2C)m^{−\alpha/n} \rightarrow 0$ but we don’t know if $\epsilon_0 \rightarrow 0$. Of course, the authors have this as a condition in Corollary 1, but when this condition is met is not known as far as I can tell. Because of this, one cannot conclude that the models are consistent. The authors mention that sometimes $\epsilon_0 = 0$ but this just shows that $\epsilon_0 = 0$ is attainable.

**Section 4.1:**

The authors state “Theorem 7 also suggests that penalizing the zero-order (C) and first-order (L) informations of the loss can improve the generalization.” This is true but it begs the question, as I mentioned above, of why? More specifically, if L becomes 0, then the loss becomes meaningless. In this case, we have that we generalize perfectly but it is not interesting. If C is decrease, why does this matter? If it is just an upper bound on the loss (as mentioned on assumption 1) then this is interesting but not intuitive. However, if the whole loss is bounded as explained in the second paragraph of Section 3, then it is clear why generalization is good, because with C = 0, the loss will always be zero and hence we generalize perfectly again but the problem is not interesting. The authors need to address these core ideas and hence this is a weakness.

**Note on theory:**

Overall the theory seems very thorough and correct, although I did not go through the proofs. For this reason, this is a strength, a really big one. The authors do a great job on the actual theory and I did not find a problem with it. Most of the paper’s weaknesses which I state above are with the writing or the interpretation of the results.

**Summary Of The Paper:**

The authors provide generalization bounds on GANs and some DNNs in terms of the Lipschitz continuity of the networks and loss functions. These bounds show that by decreasing the Lipschitz constant of the networks and loss functions, one can generalize better. Moreover, they show that with dropout and spectral normalization, one can escape the curse of dimensionality in terms of generalization error. Finally, they claim that this theory supports the empirical results we see when GANs which use Lipschitz penalization (e.g., WGAN, GP-WGAN, SN-GAN, etc.) perform well in practice.

**Summary Of The Review:**

Please provide a short summary justifying your recommendation of the paper.

**Strengths:**

1) The math is solid and seems well developed.
2) The results are interesting as they tie generalization to Lipschitz continuity.

**Weaknesses:**

1) The abstract and introduction are poorly written
2) The implications of the mathematical results have serious potential logical problems. I asked the authors to clarify in case there is some confusion. However, as it stands, it seems to me that their limiting-Lipschitzness idea does not actually reflect good generalization in practice, which is what they claim. This is because in their framework, one can use the 0 function for generator and loss and “generalize perfectly” although this is not intuitively what we want.
3) The consistency results are also questionable because they have a hypothesis in the theorems that is almost as strong as the result, which causes an almost circular reasoning. To be blunt, the actual logic is correct, but their result is very weak given this hypothesis.

If the authors had addressed these two weaknesses, I would have given them an 8 since this paper is solid otherwise.

---

> ### Author Response · Authors · 2021-11-20
> **Responses to Reviewer wjLb (part 4)**
>
> ## For some other comments:
>
> ### Lack of specificity:
>
> Both abstract and introduction parts have been revised carefully. In the new revision, we present clearly three long-standing questions for DNNs and GANs, provide some key limitations of related work.
>
> ### Some confusing points:
>
> **Notation of $z$**
>
> Thank you for pointing this out. We replaced $z$ by $x$ in Section 3 to avoid confusion.
>
> **The theorems and lemmas are labeled together**
>
> We reindexed the lemmas.
>
> **Theorem 1:**
>
> *It doesn’t seem reasonable that the loss is bounded, as most losses are not. The typical GAN loss stated in this paper is not bounded and hence Theorem 1 does not apply.*
>
> When learning a bounded function (e.g. a classifier), the assumption will satisfy if choosing a suitable loss (e.g. square loss, hinge loss, ramp loss, logistic loss, tangent loss, pinball loss) and family $H$ which has Lipschitz members with bounded ouputs. Cross-entropy loss can satisfy if we further require every $h \in H$ to have outputs belonging to a closed interval in (0,1) or use label smoothing. Note that any continuous function will be bounded in any compact subset (e.g., Z) of its domain. Therefore, we believe that the assumption of bounded loss is reasonable and applies to wide contexts. It is also worth observing that this assumption is very modest and widely used in prior theoretical studies.
>
> For GANs, we had a paragraph just below Assumption 3 to discuss the naturalness of our assumptions. Our assumptions are reasonable and  satisfied by various GANs. For example, WGAN, LSGAN, EBGAN naturally satisfy Assumption 1, while saturating GAN will satisfy it if we constraint the output of $D$ to be in $[\alpha, \beta] \subset (0, 1)$ as often used in practice. Spectral normalization and gradient penalty are popular techniques to regularize $D$ and are crucial for large-scale generators (Zhang et al., 2019; Karras et al., 2020b). Therefore Assumptions 3 and 2 are natural.
> For a comparison, existing theoretical studies (Arora et al., 2017; Zhang et al., 2018; Jiang et al., 2019; Husain et al., 2019) require this and some other stronger assumptions. (Wu et al., 2019) even assume that the training algorithm is stable, which is unrealistic for GANs.
>
> Those discussions demonstrate that our assumptions are naturally satisfied in wide contexts.
>
> *The result that you can generalize better with simpler functions seems to go against the intuition that a highly complex hypothesis class is beneficial*
>
> In our view, Theorem 1 does not contradict your intuition. Consider an overparameterized NN family $H$ with a very high capacity. Some regularization methods can help us localize a subset $H_g$ of the chosen NN family so that $H_g$ has a small generalization error. For example, in Theorem 3,  we originally need to work with family $H =${ $h_{W} :  || W_i || \le b_i$}, but Dropout localizes a subset $H_{dr} \subset H$. One should ensure that $H_{dr}$ still has a high capacity to produce a small optimization error, by choosing an appropriate drop rate.
>
> In our work we also show that deeper NNs can be better since the generalization gap can be smaller. This point is discussed in the two paragraphs before Theorem 3 in our new revision.
>
> *This is a weakness until it is explained.*
>
> Explained for Weakness \#2.
>
> **Theorem 2** might be better renamed as a proposition
>
> Despite of having a simple proof, this theorem contains a connection that leads to significant results for DNNs (and GANs) and answers two long-standing issues in deep learning. We decided to keep it as a theorem.
>
> **Section 3.2**
>
> The assumption about optimization error $\epsilon_o$ was explained before in Weakness \#3.
>
> **Section 4.1**
>
> Explained for Weakness \#2.

---

> ### Author Response · Authors · 2021-11-20
> **Responses to Reviewer wjLb (part 3)**
>
> ### 3. The consistency results are also questionable because they have a hypothesis in the theorems that is almost as strong as the result, which causes an almost circular reasoning. To be blunt, the actual logic is correct, but their result is very weak given this hypothesis.
>
> We totally agree with the reviewer that the assumption about optimization error is strong in general. However, we must respectfully disagree with the reviewer about weakness of consistency results.
>
> The assumption about optimization error $\epsilon_o(m)$ (which will decrease as $m$ increases) is naturally satisfied when the training problem is convex. Indeed, it is well-known (Allen-Zhu, 2017; Schmidt et al., 2017) that gradient descent (GD) with $T$ iterations can find a solution with error $O(T^{-1})$ whereas stochastic gradient descent (SGD) with $T$ iterations can find a solution with error $O(T^{-0.5})$. Therefore, GD and SGD with $T = O(m)$ iterations will satisfy this assumption. Note that convex training problems appear in many traditional models (Hastie et al., 2017), e.g., linear regression, support vector machines, kernel regression.
>
> For DNNs, the training problems are often nonconvex and hence the assumption may not always hold. Surprisingly, overparameterized models can lead to  tractable training problems. Indeed, (Allen-Zhu et al., 2019; Du et al., 2019; Zou et al., 2020; Nguyen \& Mondelli, 2020; Nguyen, 2021) show that GD and SGD can find global solutions of the training problems for popular DNN families. For $T$ iterations, GD and SGD can find a solution with error $O(T^{-0.5})$. Those results suggests that $T = O(m)$ iterations are sufficient to ensure our assumption about $\epsilon_o(m)$. (Allen-Zhu et al., 2019) show that $T = O(\log m)$ iterations are sufficient to ensure $\epsilon_o(m) = O(m^{-1})$.
>
> For the above reasons, we can see that our assumption about optimization error $\epsilon_o(m)$ holds in wide contexts. Furthermore, a small (even zero) optimization error is frequently observed in practice for overparameterized NNs (Zhang et al., 2021; Fang et al., 2021). In other words, our assumption is naturally satisfied for overparameterized NNs which are prevalent in practice.
>
> In the initial submission, we did discuss this point in the paragraph just below Corollary 1 and the last paragraph of Section 3. However, those discussions seem not to be focus and clear. For the new revision, we added two paragraphs after Corollary 1 to discuss the assumption clearly.

---

> ### Author Response · Authors · 2021-11-20
> **Responses to Reviewer wjLb (part 2)**
>
> (continue the answer for Weakness \#2)
>
> * Those observations suggest that making only optimization error or generalization gap small  is not enough to guarantee/explain a high performance. However, there is a large body of existing studies (Arora et al., 2021; Mianjy \& Arora, 2020; Mou et al., 2018) about generalization gap for shallow neural networks with at most one hidden layer. It has long been theoretically unknown _why deeper NNs can generalize better?_ Many great successes of deep learning often need huge datasets, but it has long been theoretically unknown _whether or not the generalization of deep NNs suffers from the curse of dimensionality?_
>
> * The above reasons suggest that we need to consider consistency. A small consistency rate is neccessary for guaranteeing a high performance. However, it is *really challenging* to guarantee a small consistency rate for truly deep NNs. Some recent works follow this direction (Kuzborskij \& Szepesvari, 2021; Ji et al., 2021; Hu et al., 2021; Jacot et al., 2018) for shallow neural networks with at most one hidden layer. However, consistency of deep neural networks remains largely open.
>
> * Note that a small consistency rate is not enough for guaranteeing a high performance. The reason is that the Bayes gap may be large in the cases of low-capacity families. When family $H$ has a too low capacity, $F(P, h^*)$ will be large and so is $F(P, h_o)$.
>
>
> _Our submission:_ Due to the reasons above, we present analyses for both generalization gap and consistency rate. Those analyses are expected to provide a big picture for the readers and provide affirmative answers for two open questions about DNNs. In the new revision, we added two new paragraphs at the end of Section 3 to discuss *sufficient condition for guaranteeing high generalization* (a small Bayes gap). This condition is novel, and basically tells that small consistency rates for high-capacity families are sufficient to guarantee high generalization.
>
> Overparameterized NNs often have a very high capacity. Some regularization methods can help us localize a subset $H_g$ of the chosen NN family so that a small generalization error for $H_g$ can be achieved. For example, in Theorem 3,  we originally need to work with family $H =${ $h_{W} :  || W_i || \le b_i$}, but Dropout localizes a subset $H_{dr} \subset H$. One should ensure that $H_{dr}$ still has a high capacity to produce a small optimization error. Interestingly, a small (even zero) optimization error is frequently observed in practice for overparameterized NNs (Zhang et al., 2021; Fang et al., 2021). In those cases, we can achieve a small consistency rate as shown in Corollary 2. This work shows such a property for Dropout and spectral normalization. We believe that many regularization methods (e.g, batch norm, group norm) have the same property. Combining these arguments with our sufficient condition will provide an answer for why those overparameterized NNs can work well on test data.
>
> We added those discussions in Appendix G of the revised version.

---

> ### Author Response · Authors · 2021-11-20
> **Responses to Reviewer wjLb (part 1)**
>
> We are happy to see that the reviewer highly appreciates our contributions in this work, and we really thank for your valuable feedbacks. The presentation of our work significantly benefited from your great suggestions/comments. We would like to take this opportunity to further provide clarifications for the remaining concerns.
>
> ## For weaknesses:
> ### 1. The abstract and introduction are poorly written
>
> We revised both carefully. In the new revision, we present clearly three long-standing questions for DNNs and GANs, provide some key limitations of related work.
>
> Those questions are:
> 1. Why Lipschitz constraint empirically can lead to great success in GANs?
> 2. Why deeper NNs can generalize better?
> 3. Whether or not the generalization of deep NNs suffers from the curse of dimensionality?
>
> ### 2. The implications of the mathematical results have serious potential logical problems. I asked the authors to clarify in case there is some confusion. However, as it stands, it seems to me that their limiting-Lipschitzness idea does not actually reflect good generalization in practice, which is what they claim. This is because in their framework, one can use the 0 function for generator and loss and “generalize perfectly” although this is not intuitively what we want.
>
> There seems to be a confusion. We also find that such a confusion may happen for other readers when only using generalization gaps in Theorem 1. Let us clarify our interpretations and then discuss our writing later.
>
> *Impractical cases:* a loss function which is constant or 0-Lipschitz continuous is not realistic, since we never use such a loss for training. On the other hand, when the learnt function is constant or 0-Lipschitz continuous, it will probably have a high training loss. So we will ignore those trivial cases in the following discussions.
>
> We often want to find an unknown (measurable) function $\eta(x)$ based on a  training set $S$ which contains $m$ i.i.d samples from distribution $P$. A popular way is to select a family $H$, a loss function $f$, and then do training on $S$ (e.g. by minimizing the empirical loss) to obtain a particular function $h_o \in H$. Our desire is that $h_o$ is as close to $\eta$ as possible. The quality of $h_o$ can be seen from different levels (Bousquet et al., 2004):
>
> - *Optimization error:* $err_o(h_o, h_m^*) =  F(\widehat{P}, h_o) - F(\widehat{P}, h_m^*)$ for comparing $h_o$ with  $h_m^* = \arg\min_{h \in H} F(\widehat{P}, h)$ which is the best in $H$ for the training data
>
> - *Generalization gap:* $err_g(h_o) =  | F(P, h_o) - F(\widehat{P}, h_o) |$ to see the difference between the empirical and expected losses of $h_o$
>
> - *Consistency rate:* $err_c(h_o, h^*) =  F(P, h_o) - F(P, h^*)$ for comparing $h_o$ with the best function $h^* = \arg\min_{h \in H} F(P, h)$ in family $H$
>
> - *Bayes gap:* $err_B(h_o, \eta) = F(P, h_o) - F(P, \eta)$ for comparing $h_o$ with the truth
>
> Obviously, a function with a small Bayes gap will have a high performance. Our desire is often to find an $h_o$ with smallest Bayes gap, meaning the best generalization ability in its family. A theory for supporting good performance of $h_o$ should show a small Bayes gap. However, it is really challenging to do so, since $\eta$ is unknown. Feasible ways for a theoretical analysis is to bound optimization error, or generalization gap, or consistency rate. We can observe that:
>
> * A small optimization error may not always lead to good generalization. Nonetheless, there is a large body of existing studies about optimization aspects, and some siginificant steps have been made recently for overparameterized models (Allen-Zhu et al., 2019; Du et al., 2019; Zou et al., 2020; Nguyen, 2021).
>
> * A small generalization gap is insufficient to explain a high success in practice. Indeed, $err_g(h_o)$ can be small although both empirical and expected losses are high. The use of uniform bounds for this quantity for neural networks (NNs) poses a long debate (Nagarajan \& Kolter, 2019; Negrea et al., 2020).
>
> * From Theorem 1, one may try to penalize the Lipschitz constant of the loss as small as possible to ensure a small generalization gap. However, as explained before, such a naive application may not lead to good performance. The reason is that family $H$ may be much smaller and its members will have lower capacity as $L$ decreases. Note that a large decrease of capacity easily leads to underfitting, and hence $F(P, h_o)$ will be high. Our experiments in Appendix F provide a further evidence when spectral normalization is overused.

---

> > ### Comment · Reviewer_wjLb · 2021-11-24
> > **More confusion from me**
> >
> > Thank you for the detailed response but I am still confused, so I will ask very specific questions to understand. My main concern is that, in Theorem 1, if we make the Lipschitz constant L of the loss too small, then we underfit (as you mentioned above). Thus, Theorem 1 does indeed imply that smaller L implies smaller generalization gap (and consistency), I agree completely. However, by making L small, we are changing the definition of what it means to generalize. To an extreme, we have $f = 0$, the impractical case that you consider above. By the same argument made in the paper we can say that our model generalizes perfectly (no generalization gap) under this scenario. But clearly that's because we are choosing a super easy loss to satisfy and does not explain good performance.
> >
> > Questions:
> > - Is anything that I stated above incorrect? If yes, please state explicitly what I said wrong so then I can understand better.
> > - If not, would you agree that your results are: smaller $L \implies$ better generalization **and not** smaller $L \implies$ better performance?

---

> > > ### Author Response · Authors · 2021-11-25
> > > **Confusion on the loss**
> > >
> > > Let us explain this point. The reviewer is confused because of allowing a change in the measuring function that defines the loss.
> > > To see this, let's write the loss for a supervised problem in details:
> > >
> > > $ f(h, x, y_x) = \psi(h(x), y_x) $
> > >
> > > where $\psi(\cdot, \cdot)$ is the measuring function that shows the difference between any given two inputs, $y_x$ is the true output, and $h(x)$ is the prediction of function $h$. Squared loss uses $\psi(y_1,y_2) = (y_1 - y_2)^2$, absolute loss uses $\psi(y_1,y_2) = |y_1 - y_2|$, hinge loss uses $\psi(y_1,y_2) = \max(0, 1 - y_1 y_2)$,...
> > >
> > > *Consider Case 1:* we change the measuring function. Instead of using $\psi$, one can use $\phi = \psi/A$ for some constant $A>0$, and hence works with a new loss $g = f/A$. He/she can make the output domain $(C)$ and Lipschitz constant $(L)$ of $g$ arbitrarily small simply by choosing $A$ to be sufficiently large. Such a loss $g$ will make all Generalization gap, Consistency rate, and Bayes gap *artificially small* !!!
> > > However, in this case, small values of $g$ simply comes from a scaling and does not reflect anything about the goodness of function/hypothesis $h$. Therefore in practice we never use such a naive scaling when training/assessing a model.
> > >
> > > *Consider Case 2:* It is possible that the Lipschitz constant $L$ of $\psi$ can be made small, by training both $\psi$ and $h$ together. In this case, $\psi$ is not fixed but can change its capacity along the training process. Obviously, this case is non-standard, since in traditional ML we often choose $\psi$ before training and fix it along the training process.
> > >
> > > We have pointed out two cases that can make a small Lipschitz constant $L$ for loss $f$. However **those cases are impractical or non-standard.**
> > >
> > > Provided that $\psi$ is chosen to be nontrivial/meaningful (as exampled before) and fixed, the only way to make small $L$ for $f$ is to choose an appropriate family $H$ and learn a suitable $h \in H$. This way is what we often do in practice. In this case, Theorem 1 does not contradict the common sense about generalization. Indeed, take $\psi$ to be squared loss, meaning $f(h,x,y_x) = (y_x - h(x))^2$. The only way to make $f(h,x,y_x) \approx 0$ at every $x$ is to ensure $h(x) \approx y_x$  at every $x$, which is what we want.
> > >
> > > **A further remark:**
> > >
> > > - The reviewer mentioned "easy loss" that may affect the bounds and indeed affect a training result significantly. It is interesting to investigate the "strength" of different losses in generalization, but may require an entirely new study.
> > > - Theorem 1 holds true for Lipschitz losses, which include both trivial and nontrivial ones. More importantly, the result for nontrivial losses is practical and meaningful.

---

### Official Review · Reviewer_NvVe · 2021-11-02

**Correctness:** 4
**Technical Novelty And Significance:** 3
**Empirical Novelty And Significance:** Not applicable
**Recommendation:** 8
**Confidence:** 3

**Main Review:**

The paper is very interesting and pretty much self-contained. It is fairly organized and connected to previous theoretical and empirical findings in the community (good related work). The background required to understand the details is provided and some quite interesting theorems are discussed and proven.
A few minor issues can be improved to make it easier to understand some sections and claims of the paper.

* in the abstract "..our bounds show that penalizing the zero- and first-order informations of the GAN loss will..." it might not be clear yet what zero or first-order information means.

* notation of Z used both for data and noise in GAN isn't a good choice.

* output of D to be in [α, β] ⊂ (0, 1) as suggested by Salimans et al. (2016). While I agree with the authors that many implementations of saturating GANs do so (e.g. α = 0.1 β=0.9), to the best of my knowledge. Salimans et al only set  β < 1 but keep α at 0.

* page 9, first line:  "one player (D or G) only can help, but maybe not enough" remove "only"?

* page 15, Definition 2: I think \bf{S} is a subset of Z^m (not an element) and ε(·) is from the power set of Z^m to R?


**Summary Of The Paper:**

This paper's primary focus is to provide insights into the relation between Lipschitz-continuity and the generalization of DNNs.
Compared to previous work, tighter bounds connecting the Lipschitz-constant of the "loss" and the expected loss of the learned function are provided. The paper also provides insights into why SN and Dropout improve generalizability in DNNs. Some quite insightful theorems connecting the generalizability of GANs to various empirical tricks commonly used to improve their performance are also provided.

**Summary Of The Review:**

I find the paper's findings quite interesting and to be the best of my verification correct.

---

> ### Author Response · Authors · 2021-11-20
> **Responses to Reviewer NvVe**
>
> We are happy to see that the reviewer highly appreciates our contributions in this work, and we really thank for your valuable feedbacks. The presentation of our work benefited much from your suggestions.
>
> Some minor comments:
>
> _1. in the abstract "..our bounds show that penalizing the zero- and first-order informations of the GAN loss will..." it might not be clear yet what zero or first-order information means._
>
> We revised the abstract carefully to reflect the problems and contributions clearer.
>
>
> _2. notation of Z used both for data and noise in GAN isn't a good choice._
>
> We replaced z by x in Section 3 to reduce confusion.
>
>
> _3. output of D to be in [α, β] ⊂ (0, 1) as suggested by Salimans et al. (2016). While I agree with the authors that many implementations of saturating GANs do so (e.g. α = 0.1 β=0.9), to the best of my knowledge. Salimans et al only set β < 1 but keep α at 0._
>
> Thank you for pointing this out. We revised it.
>
>
> _4. page 9, first line: "one player (D or G) only can help, but maybe not enough" remove "only"?_
>
> Removed. Thank you.
>
>
> _5. page 15, Definition 2: I think \bf{S} is a subset of Z^m (not an element) and ε(·) is from the power set of Z^m to R?_
>
> $Z^m$ denotes $Z \times Z \cdots \times Z$. Therefore, $\mathbf{S}$ of size $m$ is a point in $Z^m$. Of course we can consider $\mathbf{S}$ as a subset with one element.

---

### Official Review · Reviewer_yK3Y · 2021-11-04

**Correctness:** 4
**Technical Novelty And Significance:** 3
**Empirical Novelty And Significance:** Not applicable
**Recommendation:** 6
**Confidence:** 3

**Main Review:**

In this work, the authors provide the generalization and consistency of Lipschitz neural networks (NNs) with spectral normalization and dropout. Consistency is basically the difference between the objectives achieved at the optimal solution and a computed solution. In order to get a meaningful generalization bound, previous results require the number of layers to be linear to the number of samples in the worse case, whereas the authors reduce the number of layers to be logarithmic in the sample size. Furthermore, the authors derived the Lipschitz-based generalization bounds for GANs, showing that imposing zero-order and first-order constraints on GAN loss could improve GAN generalization.

Overall I think this is a work with very interesting theoretical results. The theoretical results are novel and the analysis seems to be solid. I have the following questions:

1. What does "curse of dimensionality" mean in this paper? When talking about the "curse of dimensionality", I thought that the dimensionality of each data sample is high, but it seems that the "curse of dimensionality" means something else in this paper. Is it the number of neural network layers?

2. In Theorem 1, 1 there is a $\lambda$ which is multiplied on $L$. Does that mean you can set an arbitrarily small $\lambda$ such that you do not need the Lipschitz continuity to ensure a function to generalize?

3. $q$ in Theorem 2 and 3 are re-used. $q$ appears in Theorem 3, 1 SN-DNNs and 2 Dropout DNNs, but have different meanings. In one line below Dropout DNNs in Theorem 3, there is one sentence “If the number of layers $K \ge -\frac{1}{2} \log_q m$”, which $q$ is this?

4. I understand in Theorem 1 that a smaller Lipschitz constant $L$ could lead to better generalization performance. But I’m also wondering, could a smaller $L$ also induce a higher  $F(P_z, h^*)$ value in Theorem 5 and 6? Does that mean adding the Lipschitz constraint to NNs is bad, i.e., reducing the network capacity such that the error is always high? Similarly, in Sec. 4, could a smaller $L_g$ for the generator reduce the power of the generator to model the real data distribution?

5. The bound in Theorem 8 is loose. Did the authors tighten the bound using spectral norm or Dropout?

6. The authors analyze the effectiveness of using Lipschitz constraints in GANs from the generalization perspective. But I don’t think simply from the generalization error is sufficient to explain the effectiveness of imposing Lipschitz constraints. I think Lipschitz constraints could make the optimization of GANs easier, preventing the gradient from vanishing and exploding, as analyzed in the WGAN-GP paper.


**Summary Of The Paper:**

This paper analyzes the generalization and the consistency of a function with Lipschitz continuity. In particular, the generalization and consistency bounds of Neural Networks (NNs) with Dropout or Spectral Normalization are derived. The number of layers
is logarithmic in sample size. Furthermore, the generalization of a GAN, whose discriminator and generator are both Lipschitz continuous, is given.


**Summary Of The Review:**

The theoretical results are novel and solid to me.

---

> ### Author Response · Authors · 2021-11-20
> **Responses to Reviewer yK3Y**
>
> We are happy to see that the reviewer highly appreciates our contributions in this work, and we really thank for your valuable feedbacks. We would like to take this opportunity to further provide clarifications for the remaining concerns.
>
> ### 1. What does "curse of dimensionality" mean in this paper? When talking about the "curse of dimensionality", I thought that the dimensionality of each data sample is high.
>
> We totally agree with you that the "curse of dimensionality" is talking about the dimensionality of each data sample. It suggests that a higher dimension will require an exponentially larger number of samples.
>
> Take one example. Theorem 1 shows a bound of $O(m^{-1/n})$ for a general family $\mathcal{H}$, where $n$ is the dimensionality of data. This suggests that we need $m = O(2^n)$ samples to achieve a good generalization bound. However, $O(m^{-0.5})$ in Theorem 3 suggests that for DNNs with Dropout or spectral norm, $m = O(poly(n))$ samples are sufficient, where $poly$ denotes some polynomial.
>
> Our notations in Sections 3 and 4 of the initial submission may lead to some confusions. In the revised version, we use $x$ to denote a data sample in both sections, while $z$ is the noise in GANs in Section 4. Since GANs require inputs from both real data $(x)$ and noise $(z)$, we need to consider dimensionality of both.
>
> #### Is it the number of neural network layers?
> No, as explained before.
>
> We have a paragraph that discusses the relation between depth $K$ and sample size $m$. This is to compare with existing studies. We find that our bounds are efficient in not only sample size but also network depth. Meanwhile, the existing bounds are not efficient in depth, and may explode fast as $K$ increases.
>
> While the existing studies did not show the benefits of depth on generalization, our bounds do. An interesting implication from Theorem 2 is that, fixing the norm bound on weight matrices, deeper networks will have smaller Lipschitz constants and hence lead to better generalization bounds. It has long been unknown _why deeper NNs can generalize better?_ Our work provides an answer.
>
> ### 2. In Theorem 1, 1 there is a $\lambda$ which is multiplied on $L$. Does that mean you can set an arbitrarily small $\lambda$ such that you do not need the Lipschitz continuity to ensure a function to generalize?
>
> No. There is a tradeoff in the choice of $\lambda$. The upper bound $L\lambda + C\sqrt{(\lceil {B}^{n_x}{\lambda}^{-{n_x}}\rceil \log 4 - 2 \log \delta)/m}$ shows that $\lambda$ appears in both terms. If $\lambda$ gets smaller, $\lceil {B}^{n_x}{\lambda}^{-{n_x}}\rceil $ will be exponentially larger and hence so is the second term. Therefore, we cannot ignore $L$.
>
> ### 3. q in Theorem 2 and 3 are re-used.
>
> Thank you for pointing this out. We revised Theorem 3 to remove this confused notation.
>
> ### 4. I understand in Theorem 1 that a smaller Lipschitz constant $L$ could lead to better generalization performance. But I’m also wondering, could a smaller $L$ also induce a higher $F(P_z, h^∗)$ value in Theorem 5 and 6? Does that mean adding the Lipschitz constraint to NNs is bad, i.e., reducing the network capacity such that the error is always high?
>
> We already discussed a tradeoff about $L$ in the paragraph just after Theorem 1. One should not naively try to make $L$ as small as possible, as it can lead to underfitting issues and $F(P_z, h^∗)$ can increase. Our experiments with spectral normalization (SN) in Appendix F indeed show that when SN is _overused_, the trained players of GANs can get underfitting and may hurt generalization.
>
> A suitable penalty on $L$ should be beneficial as it can reduce the generalization bounds. The consistency results in Section 3.2 suggest that we should ensure both optimization error and generalization gap to be small. Some regularization methods can help us localize a subset of the NN family so that a small generalization error can be achieved, as shown in Theorem 3 for Dropout and SN. In those cases, we can achieve a small consistency rate, meaning the test error can be small.
>
> We also find that a small consistency rate may not be enough to ensure high performance. In the new revision, we added two paragraphs just before Section 4 to provide further explanations and suggestions for high-capacity NNs.
>
> The same arguments can be used for the generator.
>
> ### 5. The bound in Theorem 8 is loose. Did the authors tighten the bound using spectral norm or Dropout?
>
> We discussed it just after Theorem 8. Combining this theorem with Theorem 2 will lead to tighter bounds. To save space, we did not present it clearly.

---

> > ### Author Response · Authors · 2021-11-20
> > **Responses to Reviewer yK3Y (2)**
> >
> > ### 6. But I don’t think simply from the generalization error is sufficient to explain the effectiveness of imposing Lipschitz constraints.
> >
> > We totally agree with you. We also discussed this point at the begining of Section 3.2. Since generalization error only is insufficient to explain the great success of DNNs, we need consistency which implicitly encloses both optimization and generalization errors. Nonetheless, the use of consistency sometimes is not sufficient. In the new revision, we added two paragraphs just before Section 4 to provide sufficient conditions for a high performance.
> >
> > 6.1. _I think Lipschitz constraints could make the optimization of GANs easier, preventing the gradient from vanishing and exploding, as analyzed in the WGAN-GP paper._
> >
> > Thank you for pointing this out. However, we think it worths an extensive study on optimization aspects. This work focuses mostly on generalization and consistency, which are complementary with optimization.

---

### Author Response · Authors · 2021-11-20
**A summary of the new revision**

Dear all reviewers and chairs,

We made the followings in the revised version:
- Abstract and Introduction were revised carefully to discuss the main problem of interests, three long-standing questions of DNNs and GANs.
- Some discussions about the assumptions of the loss and optimization error were added in Section 3.
- A "sufficient condition for guaranteeing high generalization" was added in the end of Section 3. This condition is novel and practical, since it fits well with overparameterized models. It also contributes significantly to explaining why overparameterized NNs perform well in practice. Finally, it suggests a practical way to theoretically guarantee high performance.
- Subsection 4.4 of the initial submission was moved to Appendix C.
- Appendix G was added to help the readers to see different levels when we want to theoretically evaluate a learnt function.

We hope that those revisions will make our paper to be easier to read, understand, and use theoretical results in practice.

The authors

---

### Decision · Program_Chairs · 2022-01-20

**Decision:**

Reject

**Comment:**

This paper proposes to analyze the generalization error of deep learning models and GANs using the Lipschitz coefficient of the model.

There was significant discrepancies in the evaluation of the paper among reviewers. While all reviewers acknowledged the interesting theoretical approach to understand generalization and the relevance to ICLR of the problem, they disagreed about the readiness level of the paper. Some concerns were expressed in terms of clarity (and the AC agrees with these), but most importantly, reviewer wKt9 pointed an important flaw in the current analysis that was not properly responded to by the reviewers (see below). In discussion, other reviewers were also concerned by this flaw, and so the AC decided to recommend a major revision of the paper taking the reviewers comments in consideration.

## Important flaw in the paper analysis (from wKt9)

Basically, Theorem 1 assumes that a loss $f(h,x)$ is $L$-Lipschitz w.r.t. input $x$ in some compact set of diameter $B$ for any $h$. The author shows that the:
$\sup_{h \in H} |E_{P} f(h,X) - E_{\hat{P}} f(h,X)|$ is upper-bounded by $L B + C \sqrt{\text{stuff}/m}$.

The concern of wKt9 is that the LHS is upper-bounded *trivially* and deterministically by the tighter $L B$ [see proof sketch next] for any distribution $P$ and $\hat{P}$ just because of the compactness of the input set and that $f$ is $L$-Lipschitz; one does not even need to include the number of samples $m$ in the analysis (thanks to the very strong assumption on $f$). The reviewer also was concerned that later (Theorem 3), the authors study ways that we can make $L$ exponentially small (which is interesting), but this has both the issues that:
1) it tells you nothing about the absolute performance of your network, as this only bounds the variation between any two distributions (indeed including the empirical and true distribution; but the fact that it also contains all distributions should indicate how loose this bound is!), and so perhaps the best empirical error one can obtained is still big
2) the current version of Theorem 3 uses a loose bound with a dependence on $m$ which was not even needed (as per the result above).

While it's true that empirically one can observe small empirical error, and thus combining this with a small Lipschitz constant would indicate good absolute performance; but the current presentation of the theory is rendered quite problematic by the above refinement, and should be corrected in a revision.

### Proof sketch:
For simplicity, I'll prove it for $P$ being a discrete distribution and $\hat{P}$ being the empirical; but I'm pretty sure you can extend it to continuous distributions as well.

Note that we have $|f(h,x) - f(h,x')| \leq L B$ for all $x, x'$ in the compact set of diameter $B$ and for all $h$.

Now $$E_{P} f(h,X) - E_{\hat{P}} f(h,X) = \sum_j \pi_j f(h, x_j') - \frac{1}{m} \sum_i f(h,x_i)$$

For each $x_i$, associate several $x_j$'s so that the total sum of their probabilities is $1/m$ (split some $\pi_j$ in multiple pieces if necessary) -- we can augment the index set for these new pieces, to obtain new probabilities $\pi_j'$ and call $I_i$ the set of associated indices to $x_i$. We have $\sum_{j \in I_i} \pi'_j = 1/m$

We thus have:
$$E_{P} f(h,X) - E_{\hat{P}} f(h,X) = \sum_i \sum_{j \in I_i} \pi'_j \left[ f(h, x'_j) - f(h,x_i) \right]$$

Thus:
$$|E_{P} f(h,X) - E_{\hat{P}} f(h,X)| \leq  \sum_i \sum_{j \in I_i} \pi'_j \left| f(h, x'_j) - f(h,x_i) \right| \leq L B$$

This is true for any $h$, so this is also true for the $\sup$, *deterministically*! QED